# The Antioxidant and Anti-Inflammatory Effects of Flavonoids from Propolis via Nrf2 and NF-κB Pathways

**DOI:** 10.3390/foods11162439

**Published:** 2022-08-13

**Authors:** Wenzhen Xu, Han Lu, Yuan Yuan, Zeyuan Deng, Liufeng Zheng, Hongyan Li

**Affiliations:** 1State Key Laboratory of Food Science and Technology, Nanchang University, Nanchang 330031, China; 2Guiyang Center for Disease Control and Prevention, Guiyang 550018, China; 3Institute for Advanced Study, Nanchang University, Nanchang 330031, China

**Keywords:** antioxidant, anti-inflammatory, propolis, flavonoids, Nrf2 pathway, NF-κB pathway

## Abstract

Accumulating evidence shows that oxidative stress and inflammation contribute to the development of cardiovascular disease. It has been suggested that propolis possesses antioxidant and anti-inflammatory activities. In this study, the antioxidant and anti-inflammatory effects of the main flavonoids of propolis (chrysin, pinocembrin, galangin, and pinobanksin) and propolis extract were researched. The results showed that the cellular ROS (Reactive oxygen species) levels, antioxidant enzymes, Nrf2 (Nuclear factor erythroid 2-related factor 2) nuclear translocation, and the expression of NQO1 (NAD(P)H:quinone oxidoreductase 1) and HO-1 (heme oxygenase 1) were regulated by different concentrations of individual flavonoids and propolis extract, which showed good antioxidant and pro-oxidant effects. For example, ROS levels were decreased; SOD and CAT activities were increased; and the expression of HO-1 protein was increased by chrysin. The results demonstrated that NO (Nitric Oxide), NOS (Nitric Oxide Synthase), and the activation of the NF-κB signaling pathway were inhibited in a dose-dependent manner by different concentrations of individual flavonoids and propolis extract. Moreover, the results revealed that the phytochemicals presented antioxidant effects at lower concentrations but pro-oxidant effects and stronger anti-inflammatory effects at higher concentrations. To maintain the balance of antioxidant and anti-inflammatory effects, it is possible that phytochemicals activate the Nrf2 pathway and inhibited the NF-κB (Nuclear factor kappa B) pathway.

## 1. Background

Cardiovascular disorders such as myocardial infarction are considered to be among the leading causes of mortality. Cardiovascular disorders produce excessive oxygen free radicals in the pathological process, and the disorders of free-radical metabolism in the state of oxidative stress are important triggers of myocardial damage [1]. One of the essential and indispensable immune-defense mechanisms of the human body is inflammation. However, persistent chronic inflammation can damage the visceral function and cause immune-system dysregulation, which can lead to a variety of chronic metabolic diseases or cancers [2,3]. 

Numerous research studies confirmed that oxidative stress and inflammation are interdependent and interrelated. The pro-inflammatory response could be enhanced by oxidative stress. For instance, it was demonstrated that the inflammatory phenotype of mice was modified via the genetic regulation of antioxidant defenses and ROS-generating enzymes [4,5]. In addition, the inflammatory process is also regulated by the oxidative damage of proteins, DNA, and nucleotides and the redox activation of protein kinases. For instance, mtROS increase the mitochondrial permeability and initiate a process known as “sterile inflammation” [6,7]. Another contribution of ROS to the inflammatory pathways includes the redox modulation of inflammatory mediators (e.g., High Mobility Group Box 1, S100 proteins, damage-associated molecular patterns) and transcriptional regulators associated with inflammatory pathways [8,9,10]. Furthermore, inflammatory cells release large amounts of ROS in the inflamed areas, which leads to excessive oxidative damage. The activation of immune cells is principally attributed to some superoxides and nitric oxide as well as oxidant hydrogen peroxide [11]. During inflammation, the generation of ROS is increased by leukocytes and monocytes, which enhances histological injury [12]. 

Additionally, oxidative stress caused by ROS was recently regarded as a possible mechanism in cardiovascular diseases. Numerous genes associated with oxidative stress are regulated by critical transcription factor Nrf2 [13]. Upon activation, Nrf2 is bound to the regulatory regions of specific antioxidant genes, increasing their expression and resulting in the cellular response to oxidant stress [14]. Once NF-κB (nuclear factor-kappa B (NF-κB)) is activated by lipopolysaccharides (LPSs), inflammatory responses are aggravated and magnified by the production of pro-inflammatory cytokines. Therefore, NF-κB plays a core role in the modulation of inflammatory and immunological responses. Oxidative stress and inflammation are regulated by transcription factors Nrf2 and NF-κB, respectively [15]. 

In recent years, studies showed that propolis has antioxidant, anti-inflammatory, anti-cancer, anti-bacterial, and hepatoprotective properties. Propolis is widely used as a functional food to promote public health and prevent chronic conditions such as atherosclerosis [16,17], type 2 diabetes mellitus [18,19], chronic kidney disease [20,21], and Alzheimer’s disease [22]. Propolis contains flavonoids that are beneficial to human’s health [23]. To a considerable extent, flavonoids demonstrate significant anti-inflammatory and antioxidant effects, and these effects are associated with different concentrations. Different concentrations of chrysin were shown to inhibit intracellular ROS levels, and ROS is a crucial factor in the development of oxidative stress [24]. Khezri et al. [25] indicated that chrysin reduced the cytotoxicity, MDA levels, and lysosomal and mitochondrial damage induced by AlP in a dose-dependent manner and increased the GSH activity induced by AlP in a dose-dependent manner at concentrations of 10, 50, and 100 μM. Galangin was observed to show few anti-inflammatory effects at low concentrations (0–5 μM) but to significantly inhibit the secretion of nitric oxide (NO) and nitric oxide synthase (NOS) and even the mRNA expression of inflammatory factors at concentrations of 20–30 μM [26]. Similarly, galangin at 15, 30, and 60 mg/kg inhibited the expression of NF-κB p65, NOS, TNF-α, and IL-1β in a dose-dependent manner [27]. 

Therefore, the fundamental mechanisms of antioxidant and anti-inflammatory activities in H9c2 cells were investigated using propolis extract and its main flavonoids at different concentrations. H9c2 cells are usually used to simulate heart disorders, myocardial diseases, cardiovascular diseases, etc. For example, H9c2 cells were used to induce oxidative stress in order to simulate septic cardiomyopathy by Jiang et al. [28]; to induce inflammatory injury and apoptosis in order to simulate myocardial dysfunction by Hao et al. [29]; to induce apoptosis in order to simulate acute myocardial infarction by Zhang et al. [30]; and to induce oxidative stress and inflammatory damage in order to simulate obesity and cardiovascular disease by Lama et al. [31].

## 2. Materials and Methods 

### 2.1. Materials and Sample Preparation

Propolis was bought from Jiangxi Nanchang Tongxinzicao Biological Engineering Co. Ltd. (Nanchang, China) and had been gathered in September 2015 in Nanchang (28°32′18″ N, 115°51′30″ E). H9c2 cells were bought from Procell Life Science&Technology Co., Ltd. (Wuhan, China); it is a clonal cell line subclonally obtained from BD1X rat embryonic heart tissue; Dulbecco’s modified Eagle’s medium (DMEM) and fetal bovine serum (FBS) were bought from Biological Industries (Shanghai, China). Penicillin–streptomycin liquid, Tris-buffered saline–Tween-20 (TBST), pinobanksin standards, and pinocembrin standards were bought from Solarbio Life Sciences (Beijing, China). Acetone, methanol, dimethyl sulfoxide (DMSO), and 30% hydrogen peroxide (H_2_O_2_) were bought from Damao Co. Ltd. (Tianjin, China). Cell Counting Kit-8, Total Superoxide Dismutase Assay Kit with WST-8, Catalase Assay Kit, Griess Reagent, Nitric Oxide Synthase Assay Kit, Dichlorodihydrofluorescein-Diacetate (DCF-DA), sodium dodecyl sulfate (SDS) loading buffer (6×), sulfate-polyacrylamide gel electrophoresis (SDS-PAGE), and ultra-enhanced chemiluminescence detection reagents were bought from Beyotime Biotechnology (Shanghai, China). Polyvinylidene fluoride (PVDF) membranes were bought from Roche Diagnostics GmbH (Mannheim, Germany). VCAM (1:3000), Nrf2 (1:1000), HO-1 (1:3000), and NQO1 (1:3000) were purchased from Abcam (Cambridge, UK). Phospho-NF-κB p65 (1:1000), IL-6 (1:1000), and Histone H3 (1:3000) were bought from Cell Signaling Technology (Waltham, MA, USA). β-actin (1:3000) was bought from Santa Cruz (Dallas, TX, USA). Horseradish-peroxidase-conjugated anti-rabbit (1:3000) and anti-mouse (1:3000) were bought from Signalway Antibody (Nanjing, China).

The preparation of propolis extract was performed as follows: Propolis was de-mixed and frozen at −18 °C for 24 h; then, it was crushed with a pulverizer and sieved through a 120-mesh sieve, and the finished powder was stored at −80 °C. Propolis powder was immersed in a ratio of 1:30 (*w*/*v*) in ethanol–water solution (80%, *v*/*v*) and ultrasonicated (GA92-II DA Ultrasonic cell grinder, China) at the power of 100 W and 20 kHz. The mixture was centrifuged (Heal Force Neofuge 15R high-speed freezing centrifuge; China) at 4200× *g* for 5 min. Subsequently, the supernatant was gathered, and the extraction of the residue was repeated four times at least under the same conditions as before. After that, the supernatants obtained in multiple extraction experiments were mixed together, condensed under vacuum at 37 °C, and finally lyophilized for further study. 

### 2.2. Coupling of HPLC-ESI-QTOF-MS/MS 

The liquid chromatography analyses were performed on an Agilent 1260 HPLC system (Agilent, Santa Clara, CA, USA). Chromatographic separation was implemented on an Agilent Eclipse XDB C18 (Agilent, Santa Clara, CA, USA) column with detection being carried out at 280 nm with the operating temperatures being kept at 35 °C. The analyses were completed with a gradient elution of methol (A) and 0.1% formic acid in purified water (B). The gradient protocol was: 22–36% A, 0–5 min; 36–52% A, 5–30 min; 52–63% A, 30–65 min; 63–70% A, 65–95 min; 70–80% A, 95–120 min; 80–22% A, 120–122 min. The injected sample volume was 5 μL, and the flow rate was 0.6 mL/min.

An Agilent 1260 HPLC (quaternary pump) system and AB Sciex TripleTOF™ 5600 mass spectrometer were used in the ESI-QTOF-MS/MS system. An electrospray ionization source was used to drive the TOF mass spectrometer. The capillary voltage was set to 4 kV; the collision voltage was set to 135 V; the drying-gas temperature was set to 350 °C; the drying-gas flow rate was set to 10 L/min; the nebulizer pressure was set to 40 psi; the collision gas was nitrogen; the collision energy was set to 30 eV; the full ionic scan mode was used; and the scan range was set to *m*/*z* = 50–2000.

### 2.3. Cell Culture and Treatment 

H9c2 cells were plated into DMEM containing inactivated 10% FBS, 100 U/mL penicillin, and 0.1 mg/mL streptomycin. The cells were incubated in an incubator at 37 °C with 5% CO_2_. Cells were seeded into appropriate dishes or plates for 24 h before being subjected to various treatments. Chrysin, pinobanksin, galagin, and pinocembrin were dissolved in DMSO as reserve solutions (10 mM). All reserve solutions were stored at −80 °C and diluted into DMEM at various ratios prior to cellular incubating.

### 2.4. Cell Counting Kit-8 Assay for Cell Viability

H9c2 cells (1 × 10^5^ cells per well) were seeded in a 96-well plate and allowed to complete from 80% to 90% confluence before being treated. Subsequently, the cells were exposed to various different treatments. Then, CCK-8 reagent (10 μL) was combined with 90 μL of DMEM to generate a working solution. After that, 100 μL of CCK-8 working solution was transferred towards each well, and the cells were additionally cultured for 1 h. Ultimately, the absorption at 450 nm was monitored with a microplate reader.

### 2.5. Production of Reactive Oxygen Species (ROS) 

H9c2 cells (1 × 10^6^ cells per well) were seeded inside a six-well plate and allowed to complete from 80% to 90% confluence before being treated. Cells were treated with standards and propolis extract for 12 h, followed by inducing with 150 μM H_2_O_2_ for 1 h, and then incubated with 5 mM DCF-DA (fluorescent probe) in DMEM at 37 °C for 15–20 min in the dark. Cells were digested with 0.02% EDTA (ratios of commonly used cell lysis solutions that are recommended in flow cytometry) and then centrifuged at 1500 rpm for 10 min to remove the supernatant. The pellets were re-suspended in 200 μL of cold PBS. Fluorescence intensity was monitored via flow cytometry (BD FACS, Becton Dickinson Co., Franklin Lakes, NJ, USA).

### 2.6. SOD and CAT Assays 

H9c2 cells (1 × 10^6^ cells per well) were seeded inside a six-well plate and allowed to complete from 80% to 90% confluence before being treated. After various treatments, upon lysis with a tissue lysis solution, the supernatant of H9c2 cells was collected to measure the SOD and CAT activities. The supernatant was then added to the corresponding 96-well plate in a volume of 50 μL. The intracellular SOD and CAT activities were determined according to the procedures of Total Superoxide Dismutase Assay Kit and Catalase Assay Kit. Finally, the absorbance at 560 nm (SOD) or 520 nm (CAT) was measured with a microplate reader (BioTek Instruments, Santa Clara, CA, USA). The supernatant was used to determine the protein concentration in the BCA protein-concentration assay kit. 

### 2.7. NO and NOS Assays

H9c2 cells (1 × 10^6^ cells per well) were seeded inside a six-well plate and allowed to complete from 80% to 90% confluence before being treated. After various treatments, Griess Reagent was used to detect the release of NO. Then, we took a 96-well plate and placed 50 μL of cell culture medium into the corresponding wells. A total of 50 μL of Griess Reagent I and 50 μL of Griess Reagent II were added to the corresponding wells. Nitric Oxide Synthase Assay Kit was allowed to equilibrate at 25 °C for 20 min. The treated 96-well plate was aspirated out of the culture solution, and 100 µL of NOS Assay Buffer was added. Another 100 µL of Assay Reaction Solution was added and gently mixed. Finally, the absorbance at 540 nm was measured with a microplate reader (BioTek Instruments, Santa Clara, CA, USA).

### 2.8. Extraction of Whole-Cell Protein, Cytosolic Protein, and Nuclear Protein 

H9c2 cells (1 × 10^6^ cells per well) were cultured in culture dishes 3 cm in diameter and were allowed to reach from 80% to 90% of confluence before treatment. After various treatments, cells were cleaned three times with cold PBS. Whole-cell proteins were extracted using a protein extraction kit (Shanghai, China). Briefly, after adding 200 µL of lysate/dish, the culture dishes were placed in an ice-box for 30 min and then centrifuged at 12,000× *g* for 10 min; finally, the supernatant was obtained as whole-cell proteins. 

Cytoplasmic Protein Extraction Reagent was included in Nucleoprotein and Cytoplasmic Protein Extraction Kit. We scraped off the treated cells with a cell scraper, centrifuged the cells, poured off the supernatant, and added 200 μL of PMSF-added Cytoplasmic Protein Extraction Reagent A per 20 μL of cell sediment. To completely suspend and distribute the cell sediment, it was violently vortexed for 5 s; then, it was subjected to an ice bath for 10–15 min. Then, 10 μL of Cell Plasma Protein Extraction Reagent B was added. It was vigorously vortexed for 5 s; then, it was subjected to an ice bath for 1 min. It was centrifuged at 12,000× *g* for 5 min at 4 °C after being violently vortexed for 5 s. The resulting supernatant was the cell pulp protein obtained via extraction. For the remaining precipitate, we completely aspirated the residual supernatant and added 50 μL of PMSF-added cell nuclear-protein extraction reagent. It was vigorously vortexed for 15–30 s to completely suspend and disperse the cell precipitate; then, it was returned to the ice bath and vigorously vortexed for another 15–30 s every 1–2 min for 30 min. Finally, it was centrifuged at 12,000× *g* for 10 min at 4 °C. The nucleoproteins recovered from the cells were observed in the supernatant.

### 2.9. Western Blot Analysis 

Proteins (about 20–30 µg) were mixed with sodium dodecyl sulfate (SDS) loading buffer (6×; Beyotime Biotechnology, Shanghai, China), and proteins were detected via 10% sodium dodecyl sulfate-polyacrylamide gel electrophoresis (SDS-PAGE) and transferred to PVDF membranes using constant current. After being blocked with 5% skimmed milk in TRIS-buffered saline–Tween-20 for 2 h at room temperature (about 25 °C), PVDF membranes were incubated with the corresponding primary antibodies VCAM (1:3000), Nrf2 (1:1000), HO-1 (1:3000), NQO1 (1:3000), phospho-NF-κB p65 (1:1000), IL-6 (1:1000), and Histone H3 (1:3000) or β-actin (1:5000) overnight at 4 °C. The membranes were washed with TBST and then incubated with horseradish-peroxidase-conjugated anti-mouse (1:3000) or anti-rabbit (1:3000) secondary antibodies in TBST for 2 h at room temperature. Finally, target bands were observed in enhanced chemiluminescence (ECL) detection solution using the ECL technique (Image Lab™ Touch Software, Bio-Rad Laboratories, Inc., Hercules, CA, USA) in the dark after three washes with TBST. The intensity of the target bands was determined using an imaging analyzer (Image Lab™ Touch Software, Bio-Rad Laboratories, Inc., Hercules, CA, USA). The relative expression of proteins was standardized to Histone H3 or β-actin. 

### 2.10. Statistical Analysis 

The SPSS 26.0 statistical software tool (SPSS Inc., Chicago, IL, USA) was used to analyze the data, which were given as mean standard errors of the mean (SEMs). Duncan’s tests were used to assess differences after a single-factor analysis of variance (ANOVA), and *p* < 0.05 was used to denote significant differences. The data were statistically analyzed using GraphPad Prism 8 software (San Diego, CA, USA).

## 3. Results 

### 3.1. Identification of Chemical Constituents in Propolis Extract

Eleven flavonoids, including chrysin, galangin, pinocembrin, pinobanksin, pinobanksin 3-O-acetate, pinobanksin-3-O-butyric, pinobanksin-5-methyl ether, apigenin-7-O-glucoside, methoxyl-cyanide, galangin-5-methyl ether, and pinocembrin-7-methyl ether, were identified in propolis extract by comparing the coupling of HPLC-ESI-QTOF-MS/MS fragmentation with reference standards and literature information [32,33,34,35] (Figure 1 and Table 1). The ESI-QTOF-MS/MS data showed that the molecular ions of Peak 5, Peak 6, Peak 7, Peak 11, Peak 14, and Peak 15 were 285.0689, 431.0854, 271.0612, 255.0663, 253.0419, and 269.0418, respectively, which corresponded with the reference standards. The characteristic fragment ion (284.0228 [M-H-CH3]^−^) of Peak 8, the characteristic fragment ion ([M-H-CO_2_-CO]^−^ = 211.0319) of Peak 16, and the characteristic fragment ion ([M-H-CH_3_-CO]^−^ = 226.0548) of Peak 17 were identified as methoxyl-cyanidenon [36]. Peaks 12 and 18 were identified as shortleaf-pine derivatives, having similar fragment ions (271.0509 [M-acetate]^−^ and 253.0419 [M-acetate-H_2_O]^−^), which may be related to the cleavage of acyl bonds (acetyl, butyryl). Peaks 12 and 18 were identified as 3-O-acetyl-pinobanksin (*m*/*z* = 313.0612) and 3-O-butyryl-pinobanksin (*m*/*z* = 341.0917), respectively, based on fragment ions, molecular ions, and the literature [37]. From the results and previous data [38], the main flavonoids of propolis extract were identical to chrysin (30.56 ± 0.60 mg/g DW), pinocembrin (30.96 ± 0.34 mg/g DW), galangin (16.69 ± 0.45 mg/g DW), and pinobanksin (9.56 ± 0.25 mg/g DW; Figure 2).

### 3.2. Effects of H_2_O_2_ on Cell Viability and Production of ROS

To induce oxidative stress, H9c2 cells were exposed to multiple H_2_O_2_ concentrations (0, 50, 100, 150, 200, and 250 μM) for 1 h. Compared with the control group, cell viability was 60.24% (Appendix A) when the concentration of H_2_O_2_ reached 150 μM. In addition, the level of intracellular ROS was significantly increased to 292.20% when the H_2_O_2_ concentration was 150 μM. Compared with the control group, ROS levels were gradually decreased to 205.12% (Appendix A) when the concentration of H_2_O_2_ reached 250 μM. ROS levels were gradually increased when the concentration of H_2_O_2_ was in the range of 0–150 μM. ROS levels were gradually decreased when the concentration was above 150 μM (*p* < 0.05). The concentration with the highest intracellular ROS level was used in subsequent experiments. Cells were in a state of oxidative stress when the viability was 50–70%, which does not immediately cause death and can be recovered using antioxidants. Oxidation significantly harms cells and does irreparable damage when cell viability falls below 40%. Therefore, 150 μM H_2_O_2_ was used to induce cellular oxidative damage in H9c2 cells. 

### 3.3. Effects of LPSs on Cell Viability and Expression of Pro-Inflammatory Cytokine Proteins

The viability of H9c2 cells was unaffected by LPS concentrations of 5, 10, 15, 20, 25, 30, 35, and 40 μg/mL (Appendix A). Compared with the control group, when the concentration of LPSs reached 10 μg/mL, the expression of VCAM and IL-6 pro-inflammatory cytokine proteins was significantly increased to 150.76% and 181.05%, respectively (Appendix A). The protein expression of VCAM and IL-6 was maximized when the concentrations of LPSs reached 10 μg/mL. Therefore, 10 μg/mL LPSs were used to model the cellular inflammatory damage.

### 3.4. Effects of Chrysin, Pinocembrin, Galangin, Pinobanksin, and Propolis Extract on Cell Viability 

The CCK-8 assay was used to assess the potential cytotoxicity of chrysin, pinocembrin, galangin, pinobanksin, and propolis extract in H9c2 cells. When the concentration of chrysin was higher than 30 μΜ, that of galangin higher than 60 μM, and that of propolis extract higher than 100 μg/mL, cell viability was below 95%. However, pinocembrin below 300 μM and pinobanksin below 300 μM did not inhibit the proliferation of H9c2 cells (Appendix A). In order to avoid toxicity, the concentration of chrysin below 30 μM, that of pinocembrin below 80 μM, that of galangin below 60 μM, that of pinobanksin below 80 μM, and that of propolis extract below 100 μg /mL were chosen for further study. 

### 3.5. Effects of Chrysin, Pinocembrin, Galangin, Pinobanksin, and Propolis Extract on Cell Production of ROS

ROS are among the major intracellular oxidation products and important participants in cell signaling. Their accumulation probably causes macrophages to undergo more apoptosis or autophagy [39]. Pretreatment with different sample concentrations dramatically reduced H_2_O_2_-induced ROS generation (*p* < 0.05), and significant differences were found among doses that showed a trend of first decreasing and then remaining constant or increasing. On the one hand, after chrysin, pinocembrin, and propolis-extract pretreatment, the antioxidant effects tended to first increase and subsequently decrease. On the other hand, after galangin and pinobanksin pretreatment, the antioxidant effects tended to first increase and subsequently stabilize.

Chrysin (Figure 3A), compared with the H_2_O_2_-induced group (373.35 ± 2.42%), showed anti-oxidant effects in the concentration range from 5 μM (361.15 ± 1.57%) to 10 μM (243.38 ± 1.22%), while it showed pro-oxidant effects in the concentration range from 15 μM (331.01 ± 4.16%) to 25 μM (354.88 ± 1.58%). Pinocembrin (Figure 3B), compared with the H_2_O_2_-induced group (373.64 ± 17.37%), showed anti-oxidant effects in the concentration range from 5 μM (294.56 ± 7.6%) to 40 μM (179.94 ± 6.2%), while it showed pro-oxidant effects in the concentration range from 60 μM to 80 μM. Galangin (Figure 3C), compared with the H_2_O_2_-induced group (430 ± 1%), showed anti-oxidant effects in the concentration range from 10 μM (384.59 ± 0.87%) to 50 μM (232.27 ± 6.69%), while it showed pro-oxidant effects in the concentration range from 50 μM to 60 μM (240.7 ± 2.33%). Pinobanksin (Figure 3D), compared with the H_2_O_2_-induced group (461.88 ± 36.48%), showed anti-oxidant effects in the concentration range from 5 μM (320.97 ± 14.04%) to 40 μM (179.94 ± 6.2%), while it showed pro-oxidant effects in the concentration range from 40 μM to 80 μM (182.81 ± 4.3%). Propolis extract (Figure 3E), compared with the H_2_O_2_-induced group (331.87 ± 4.56%), showed anti-oxidant effects in the concentration range from 10 μg/mL (236.43 ± 3.16%) to 40 μg/mL (49.78 ± 2.6%), while it showed pro-oxidant effects in the concentration range from 60 μg/mL (112.79 ± 1.58%) to 100 μg/mL (223.64 ± 3.17%).

### 3.6. SOD and CAT Activities

SOD plays an essential role in the conversion of superoxide to hydrogen peroxide, while CAT converts hydrogen peroxide to water [40]. To explore the effect of the antioxidant defense of chrysin, pinocembrin, galangin, pinobanksin, and propolis extract, SOD (Figure 4) and CAT (Figure 5) activities were evaluated. Compared with the control group, these enzyme activities were significantly decreased in the H_2_O_2_-induced group (*p* < 0.05). Furthermore, after pretreatment with different sample concentrations, SOD and CAT levels were increased in H9c2 cells induced by H_2_O_2_ (*p* < 0.05). For example, as shown in Figure 4A and Figure 5A, compared with the H_2_O_2_ group (SOD, 287.55 ± 1.07 U/mg protein; CAT, 1.43 ± 0.12 U/mg protein), SOD activity (296.97 ± 2.23–298.56 ± 1.59 U/mg protein) and CAT activity (2.06 ± 0.11–3.04 ± 0.05 U/mg protein) were increased after 5–25 μM chrysin pretreatment.

### 3.7. Effects of Chrysin, Pinocembrin, Galangin, Pinobanksin, and Propolis Extract on the Expression of Proteins Encoded by Antioxidant Genes Downstream of Nrf2

A number of antioxidant genes are activated by nuclear factor erythroid 2-related factor 2 (Nrf2) to protect the body from ROS damage [41,42]. Upon exposure to ROS, Nrf2 is translocated to the nucleus and conjugated to antioxidant response elements (ARE); thereby, the transcription of cellular-defense-related genes is up-regulated, including antioxidant proteins, antitoxic enzymes, and drug metabolism [42,43]. Therefore, the expression of antioxidant proteins downstream of Nrf2, HO-1 and NQO-1, was assayed via Western blot. Compared with the control group, the expression of HO-1 and NQO-1 proteins was significantly decreased in H_2_O_2_-induced H9c2 cells (*p* < 0.05). In addition, the expression of HO-1 and NQO-1 proteins was dramatically increased after pretreatment with different sample concentrations in H_2_O_2_-induced H9c2 cells (*p* < 0.05; Figure 6).

On the one hand, after pretreatment with chrysin, pinocembrin, and propolis extract, the expression of the HO-1 protein tended to first increase and subsequently decrease. For example, with chrysin (Figure 6A), compared with the H_2_O_2_-induced group (80.73 ± 0.88%), the expression of HO-1 was significantly increased, and the antioxidant effect increased in the concentration range from 5 μM (98.85 ± 3.22%) to 15 μM (95.31 ± 1.02%), while the expression of HO-1 was gradually decreased and showed pro-oxidant effects in the concentration range from 20 μM (78.69 ± 9.44%) to 25 μM (62.59 ± 0.06%). On the other hand, after galangin and pinobanksin pretreatment, the expression of the HO-1 protein tended to first increase and subsequently stabilize. For example, with pinobanksin (Figure 6D), compared with the H_2_O_2_-induced group (46.7 ± 0.74%), the expression of HO-1 was significantly increased, and the antioxidant effect increased in the concentration range from 5 μM (75.13 ± 0.83%) to 40 μM (96.85 ± 3.82%), while the expression of HO-1 was no longer consistently elevated and showed pro-oxidant effects in the concentration range from 40 μM to 80 μM (105.95 ± 4%).

Compared to the H_2_O_2_-induced group, there were no significant differences in the expression of NQO1 after chrysin pretreatments (*p* > 0.05). However, after galangin, pinobanksin, and propolis-extract pretreatments, the expression of the NQO1 protein tended to first increase and then stabilize. For example, with propolis extract (Figure 6E), compared with the H_2_O_2_-induced group (60.36 ± 0.38%), the expression of NQO1 was significantly increased, and the antioxidant effect increased in the concentration range from 5 μM (90.88 ± 4.4%) to 40 μM (106.87 ± 7.5%), while the expression of the NQO1 protein was no longer consistently elevated and showed pro-oxidant effects in the concentration range from 40 μM to 100 μM (114.55 ± 1.97%). In addition, with pinocembrin (Figure 6B), compared with H_2_O_2_-induced group (82.59 ± 0.84%), in the concentration range from 5 μM (90.50 ± 2.3%) to 80 μM (107.87 ± 0.37%), the expression of the NQO1 protein was significantly increased.

### 3.8. Activation of Nrf2 by Chrysin, Pinocembrin, Galangin, Pinobanksin, and Propolis Extract

Nrf2 nuclear translocation in H9c2 cells was detected via Western blot. Compared with the control group, the expression of nuclear Nrf2 was significantly decreased in the H_2_O_2_-induced group (*p* < 0.05). After treatment with different concentrations of chrysin (Figure 7A), pinocembrin (Figure 7B), galangin (Figure 7C), pinobanksin (Figure 7D), and propolis extract (Figure 7E), the translocation of Nrf2 from the cytoplasm to the nucleus was enhanced.

On the one hand, after pretreatment with different concentrations of chrysin, pinocembrin, and propolis extract, the accumulation of Nrf2 in the nucleus tended to first increase and subsequently decrease as the concentration was gradually increased. For example, with chrysin (Figure 7A), compared with the H_2_O_2_-induced group, the translocation of Nrf2 from the cytoplasm to the nucleus was gradually increased and showed anti-oxidant effects in the concentration range from 5 to 10 μM, while the translocation of Nrf2 from the cytoplasm to the nucleus was gradually decreased in a dose-dependent manner and showed pro-oxidant effects in the concentration range from 15 μM to 25 μM.

On the other hand, after pretreatment with different concentrations of galangin and pinobanksin, the accumulation of Nrf2 in the nucleus tended to first increase and subsequently stabilize as the concentration was gradually increased. For example, with galangin, compared with the H_2_O_2_-induced group, the translocation of Nrf2 from the cytoplasm to the nucleus was increased in a dose-dependent manner and showed anti-oxidant effects in the concentration range from 10 μM to 50 μM, while the translocation of Nrf2 from the cytoplasm to the nucleus reached saturation and showed pro-oxidant effects in the concentration range from 50 μM to 60 μM.

It is worth noting that the increase in cytoplasmic Nrf2 was always accompanied by the decrease in nuclear Nrf2. Additionally, the expression of the nuclear translocation of Nrf2 followed a tendency similar to that of the expression of the HO-1 protein. According to these results, the modification of Nrf2 nuclear translocation was the mechanism via which chrysin, pinocembrin, galangin, pinobanksin, and propolis extract showed antioxidant activity. It was found that Nrf2 translocation from the cytoplasm to the nucleus was up-regulated (chrysin range of 5 μM–10 μM, pinocembrin range of 5 μM–40 μM, and propolis-extract range of 5 μg/mL–40 μg/mL) and then down-regulated (chrysin range of 15 μM–25 μM, pinocembrin range of 40 μM–60 μM, and propolis-extract range of 40 μg/mL–100 μg/mL) following treatments with chrysin, pinocembrin, and propolis extract. It was also found that Nrf2 translocation from the cytoplasm to the nucleus was up-regulated and then held relatively constant following treatments with 10–60 μM galangin and 5–80 μM pinobanksin.

### 3.9. NO and NOS Levels

NO is catalyzed by NOS in cells, and it can trigger tissue damage and ultimately lead to pain and inflammation [44,45]. The secretion of NO was significantly reduced by chrysin, pinocembrin, galangin, pinobanksin, and propolis extract (*p* < 0.05; Figure 8). For example, compared with the control group (2.95 ± 0.1 μM), the levels of NO (4.98 ± 0.13 μM) were significantly increased in H9c2 cells in the LPS-induced group. Moreover, the levels of NO were significantly reduced (1.866 ± 0.03 μM–4.69 ± 0.03 μM) by 5–25 μM chrysin.

In addition, NOS generates NO, which is an inflammatory mediator [46]. The levels of NOS were considerably higher in the treatment group than that in the control group in H9c2 cells induced with 10 µg/mL LPS, and the levels of NOS were significantly attenuated by chrysin, pinocembrin, galangin, pinobanksin, and propolis extract in a dose-dependent manner (*p* < 0.05; Figure 9). For example, compared with the control group (100%), the level of NO (120 ± 14.65%) was significantly increased in H9c2 cells induced with 10 µg/mL LPS. Moreover, the level of NO was significantly decreased (82.13 ± 2.7%–90.09 ± 0.6%) after 5–25 μM chrysin pretreatment in H9c2 cells induced with 10 µg/mL LPS.

### 3.10. Chrysin, Pinocembrin, Galangin, Pinobanksin, and Propolis Extract Down-Regulated the Expression of Pro-Inflammatory Cytokines

To explore the effects of chrysin, pinocembrin, galangin, pinobanksin, and propolis extract on the inflammatory response of LPS-induced H9c2 cells, the expression of pro-inflammatory cytokines was measured. Compared with the control group, the expression of IL-6 (109.63–230.75%) and VACM proteins (*p* < 0.05; 167.29–245.79%) was significantly up-regulated in the LPS-induced group. After pretreatment with the samples, the expression of IL-6 and VCAM proteins was significantly inhibited in LPS-induced H9c2 cells in a dose-dependent manner (*p* < 0.05; Figure 10).

### 3.11. Inhibition of NF-κB Signaling Pathway

The NF-κB signaling pathway can be activated by the up-regulation of pro-inflammatory cytokines [47]. After pretreatment with different concentrations of chrysin, pinocembrin, galangin, pinobanksin, and propolis extract, the expression of the phosphorylation of the NF-κB p65 protein was examined to explore the possible mechanism of inhibition of the expression of VCAM1 and IL-6 as well as to confirm the effects on the NF-κB signaling pathway. Compared with the control group, the expression of the phosphorylation of NF-κB p65 was increased in the LPS-induced group. However, after pretreatment with the samples, the expression of the phosphorylation of NF-κB p65 was significantly attenuated in a dose-dependent manner (*p* < 0.05; Figure 11). For example, with chrysin (Figure 11A), compared with the LPS-induced group (138.49 ± 10.2%), the expression of the p65 protein was decreased to 212.11 ± 27.49% when the concentration of chrysin was 5 μM. The expression of the p65 protein gradually decreased to 121.05 ± 18.9% when the concentration of chrysin was gradually increased to 25 μM.

The results indicated that chrysin, pinocembrin, galangin, pinobanksin, and propolis extract markedly suppressed the phosphorylation of NF-κB p65 induced by LPS.

## 4. Discussion

### 4.1. Interpretation of Antioxidant and Pro-Oxidant Effects

In terms of structure, flavonoid compounds provide hydrogen atoms to bind to oxygen radicals and prevent the formation of free radicals. Additionally, the signaling pathways associated with antioxidant defense systems are regulated by flavonoid compounds. This could be how flavonoids interact to exert their antioxidant effects.

Firstly, the structure of a flavonoid indicates whether it has pro- or antioxidant effects. The structures of chrysin, pinocembrin, galangin, and pinobanksin are shown in Figure 2. It can be observed that chrysin, pinocembrin, and pinobanksin are flavonoids, whereas galangin is a flavonol compound. Because of the C_2_-C_3_ double bond and the C-3 hydroxyl group, flavonols (e.g., galangin) have a stronger oxidative activity, while flavonoids have a lower oxidative activity [48,49]. It is worth emphasizing that the C_2_=C_3_ double bond plays an important role in the antioxidant activity of flavonoids. Chrysin and galangin both feature C_2_=C_3_ double bonds, but galangin differs in that it has a C-3 hydroxyl group that promotes the oxidation to benzoquinone intermediates. However, the oxidation of galangin requires the consumption of free radicals in the system. Due to the lack of a B-ring ortho-hydroxyl group, the pro-oxidation effect of galangin is weaker than that of chrysin [50]. Therefore, a higher concentration of galangin does not show pro-oxidation effects.

Secondly, chrysin, pinocembrin, galangin, pinobanksin, and propolis extract exhibited antioxidant and pro-oxidant effects in a dose-dependent manner. The occurrence of pro-oxidant effects may be due to the auto-oxidation of flavonoid compounds, because as the concentration of these flavonoids increases, so does the synthesis of lipid peroxidation products and the development of superoxide anion radicals [51]. Moreover, once flavonoids reach a higher concentration, they may be oxidized to intermediates with pro-oxidant effects such as phenoxy radicals, semi-quinones, and quinone structures [52]. Therefore, high concentrations of chrysin, pinocembrin, and propolis extract showed pro-oxidant effects. Similarly, to confirm the pro-oxidant effect of high doses of genistein (200 μM) in primary muscle cells, Chen et al. [53] analyzed cellular lipid peroxidation, redox homeostasis, and ROS production. Furthermore, Galati et al. [54] indicated that 3 mM epigallocatechin and 2 mM epicatechin-3-gallate were found to have pro-oxidant effects on the mitochondrial-membrane potential and ROS levels in hepatocytes.

### 4.2. Activation of Nrf2 Signaling Pathway and Inhibition of NF-κB Signaling Pathway

The individual flavonoids (chrysin, pinocembrin, galangin, and pinobanksin) and propolis extract exerted strong antioxidant effects as inducers of the Nrf2/HO-1 axis, and they were potent activators of Nrf2 nuclear translocation in H9c2 cells. The phytochemicals showed antioxidant effects at low concentrations and showed pro-oxidant effects at high concentrations, possibly by activating Nrf2. Once activated by oxidative stress, Nrf2 is translocated to the nucleus and binds to antioxidant transcription elements in the promoter region of phase 2 to increase the expression of certain antioxidant and detoxification genes, ultimately leading to cellular resistance to oxidative stress [55]. HO-1 and NQO-1 are well-characterized Nrf2-dependent antioxidant defense genes. It was speculated that transcription factor Nrf2, which was associated with the degree of SOD and CAT activation of cellular antioxidant responses, was enhanced or lowered by chrysin, pinocembrin, and propolis extract in the nucleus of H9c2 cells. Intriguingly, the levels of antioxidant enzymes were typically associated with amplification during the formation of ROS. SOD levels represent a fundamental defense mechanism against excessive ROS [56]. Similar results were observed in quercetin that pretreating with low concentrations of flavonoid stimulated cell proliferation and enhanced the total antioxidant capacity of cells. Moreover, higher concentrations of the flavonoid diminished cell viability and total antioxidant capacity, as well as the activities of catalase, superoxide dismutase, and glutathione S-transferase [57].

Oxidative stress and inflammatory responses are important components in the pathogenesis of cardiovascular stent disease. Accumulating evidence suggests that oxidative stress is inseparable from the inflammatory response. Currently, exploring the relationship between antioxidant and anti-inflammatory agents is a potential target for the prevention or mitigation of cardiovascular disease. Therefore, our study focused on the changes in the Nrf2 and NF-κB pathways involved in oxidative stress and inflammatory processes.

According to the datas of ROS, NOS, and the expression of pro-inflammatory protein, chrysin showed anti-oxidant effects in the concentration range from 5 μM to 10 μM, while it showed pro-oxidant effects in the concentration range from 15 μM to 25 μM, and it showed anti-inflammatory effects in the concentration range from 5 μM to 25 μM. Pinocembrin showed anti-oxidant effects in the concentration range from 5 μM to 40 μM, while it showed pro-oxidant effects in the concentration range from 60 μM to 80 μM, and it showed anti-inflammatory effects in the concentration range from 5 μM to 80 μM. Galangin showed anti-oxidant effects in the concentration range from 10 μM to 50 μM, while it showed pro-oxidant effects in the concentration range from 50 μM to 60 μM, and it showed anti-inflammatory effects in the concentration range from 10 μM to 60 μM. Pinobanksin showed anti-oxidant effects in the concentration range from 5 μM to 60 μM, while it showed pro-oxidant effects in the concentration range from 60 μM to 80 μM, and it showed anti-inflammatory effects in the concentration range from 5 μM to 80 μM. Propolis extract showed anti-oxidant effects in the concentration range from 5 μg/mL to 40 μg/mL, while it showed pro-oxidant effects in the concentration range from 60 μg/mL to 100 μg/mL, and it showed anti-inflammatory effects in the concentration range from 5 μg/mL to 100 μg/mL. Additionally, the higher the concentration of a compound was, the stronger the anti-inflammatory effect was.

The inhibition of the production of NOS induced by flavonoids is caused by their antioxidant properties, and these compounds can exert anti-inflammatory effects by scavenging ROS [58]. Furthermore, flavonoids are used as inhibitors of lipopolysaccharide-signaling molecules to decrease inflammation.

NF-κB is a classical inflammatory signaling pathway that regulates the expression of immune genes encoding cytokines, such as IL-1β, TNF-α, and IL-6 [47]. Previous reports showed that propolis inhibited cytokine production in various cardiovascular injuries [59]. Likewise, it was also observed that propolis inhibited the expression of VCAM1 and IL-6 in our findings. It was documented that NOS, VCAM, and IL-6 were pivotal for NF-κB signaling [60]. The anti-inflammatory effects were enhanced by chrysin, pinocembrin, galangin, pinobanksin, and propolis-extract pretreatment with the increase in concentration. Moreover, the inhibition of the NF-kB signaling pathway was also enhanced. It was speculated that the pro-oxidant effects were produced by high concentrations of phytochemicals triggering a series of inflammatory responses and exerting anti-inflammatory effects by subsequently inhibiting the NF-κB signaling pathway. It was in accordance with the trend of progressively lower expression of inflammatory proteins VCAM and IL-6 downstream of the NF-κB signaling pathway in the results.

Research demonstrates that a functional interaction and crosstalk between Nrf2 and NF-κB pathways exists to maintain balance or regulate oxidative stress and inflammation [15]. Even though many phytochemicals are reported to modulate NF-κB and Nrf2 activities [61], the mechanism of crosstalk remains unclear. Therefore, it is possible that the activation of the Nrf2 pathway dominates at low phytochemical concentrations and that the inhibition of the NF-κB pathway dominates at high phytochemical concentrations.

### 4.3. Bioavailability Issue of Chrysin, Pinocembrin, Galangin, Pinobanksin, and Propolis Extract

A total of 18 substances were identified in the ethanolic extract of propolis, including flavonoids and their derivatives, and phenolic acids and their esters. According to previous data, the main flavonoids of propolis extract were identical to chrysin (30.56 ± 0.60 mg/g DW), pinocembrin (30.96 ± 0.34 mg/g DW), galangin (16.69 ± 0.45 mg/g DW), and pinobanksin (9.36 ± 0.28 mg/g DW).

The bioavailability values of galangin and chrysin in propolis extracts were determined in a study, and they were at 7.8% and 7.5%, respectively [62]. Moreover, propolis extract has a higher bioavailability than single-flavonoid standards [32]. In addition, galangin is frequently used with popular pharmaceuticals. So, it has the potential to improve the bioavailability and chemoprevention of oral drugs and to reverse multidrug resistance [63]. Furthermore, there are numerous ways to improve bioavailability, which sparked renewed interest in propolis research. The above findings make propolis extract a promising antioxidant for use as a food supplement.

## 5. Conclusions

The major flavonoids in propolis were identified as chrysin, pinocembrin, galangin, and pinobanksin. It was revealed that flavonoids from propolis mainly presented antioxidant effects at lower concentrations and presented pro-oxidant as well as stronger anti-inflammatory effects at higher concentrations. We observed anti-oxidant effects of chrysin in the concentration range from 5 μM to 10 μM, of pinocembrin in the concentration range from 5 μM to 40 μM, of galangin in the concentration range from 10 μM to 50 μM, of pinobanksin in the concentration range from 5 μM to 60 μM, and of propolis extract in the concentration range from 5 μg/mL to 40 μg/mL, while we observed pro-oxidant effects of chrysin in the concentration range from 15 μM to 25 μM, of pinocembrin in the concentration range from 60 μM to 80 μM, of galangin in the concentration range from 50 μM to 60 μM, of pinobanksin in the concentration range from 60 μM to 80 μM, and of propolis extract in the concentration range from 60 μg/mL to 100 μg/mL. In addition, we observed anti-inflammatory effects of chrysin in the concentration range from 5 μM to 25 μM, of pinocembrin in the concentration range from 5 μM to 80 μM, of galangin in the concentration range from 10 μM to 60 μM, of pinobanksin in the concentration range from 5 μM to 80 μM, and of propolis extract in the concentration range from 5 μg/mL to 100 μg/mL. Additionally, the higher the concentration of the compound was, the stronger the anti-inflammatory activity was. Flavonoids from propolis could probably activate the Nrf2 pathway and inhibit the NF-κB pathway to maintain the balance of antioxidant and anti-inflammatory effects. In the future, it is important to focus on the link between the Nrf2 and NF-κB pathways to explore the anti-oxidative mechanism of low-concentration flavonoids and the anti-inflammatory mechanism of high-concentration flavonoids.

## Figures and Tables

**Figure 1 foods-11-02439-f001:**
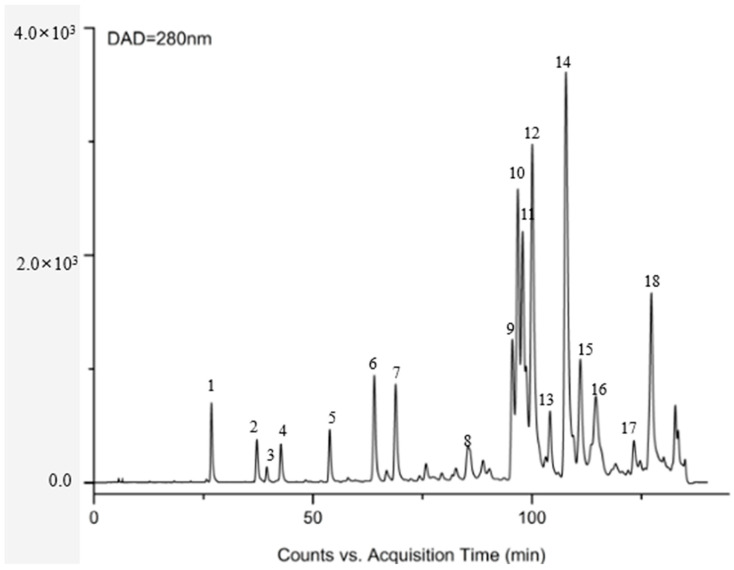
DAD chromatogram at 280 nm of propolis.

**Figure 2 foods-11-02439-f002:**
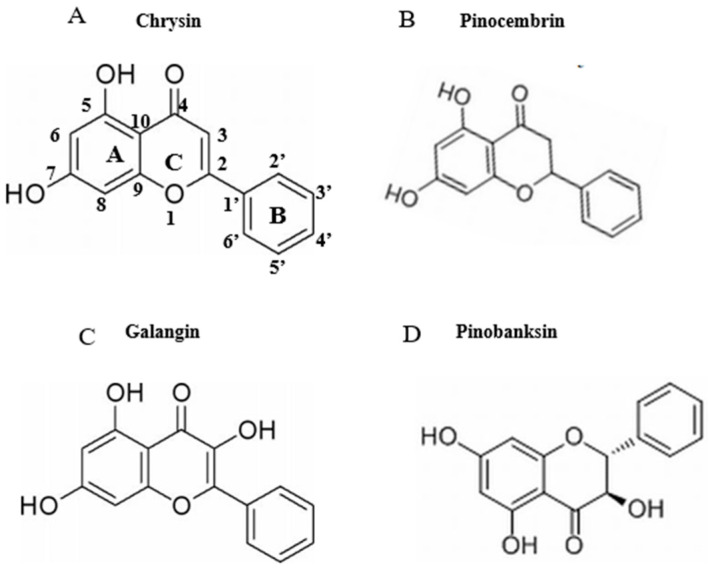
Chemical structures of flavonoids (chrysin, pinocembrin, galangin, and pinobanksin). (**A**) chrysin; (**B**) pinocembrin; (**C**) galangin; (**D**) pinobanksin.

**Figure 3 foods-11-02439-f003:**
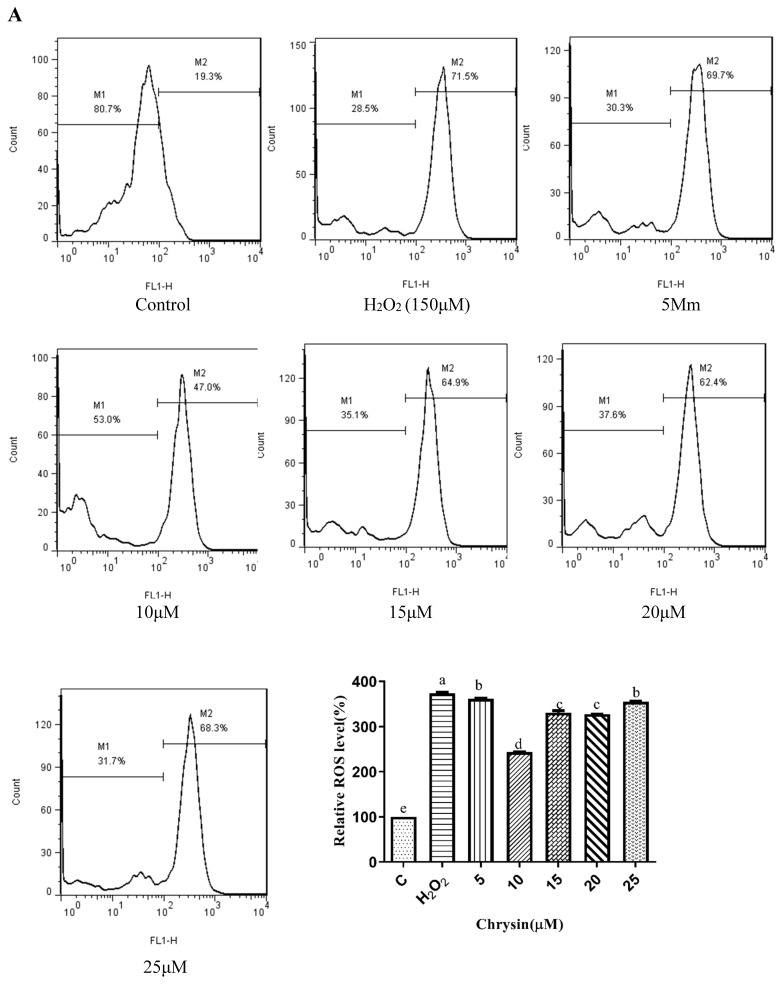
ROS levels induced by individual flavonoids (chrysin, pinocembrin, galangin, and pinobanksin) and propolis extract in 150 μM H_2_O_2_-induced H9c2 cells. H9c2 cells were stimulated with different concentrations of phytochemicals (chrysin, pinocembrin, galangin, pinobanksin, and propolis extract) for 12 h and then treated with 150 μM H_2_O_2_ for 1 h; finally, ROS levels were detected using a flow-cytometry assay. (**A**) ROS levels in chrysin-induced H9c2 cells. (**B**) ROS levels in pinocembrin-induced H9c2 cells. (**C**) ROS levels in galangin-induced H9c2 cells. (**D**) ROS levels in pinobanksin-induced H9c2 cells. (**E**) ROS levels in propolis-extract-induced H9c2 cells. Values are expressed as the mean ± SEM (*n* = 3). Values with different letters (a, b, c, d, e, f, g, h) in the figure showed significant differences at *p* < 0.05.

**Figure 4 foods-11-02439-f004:**
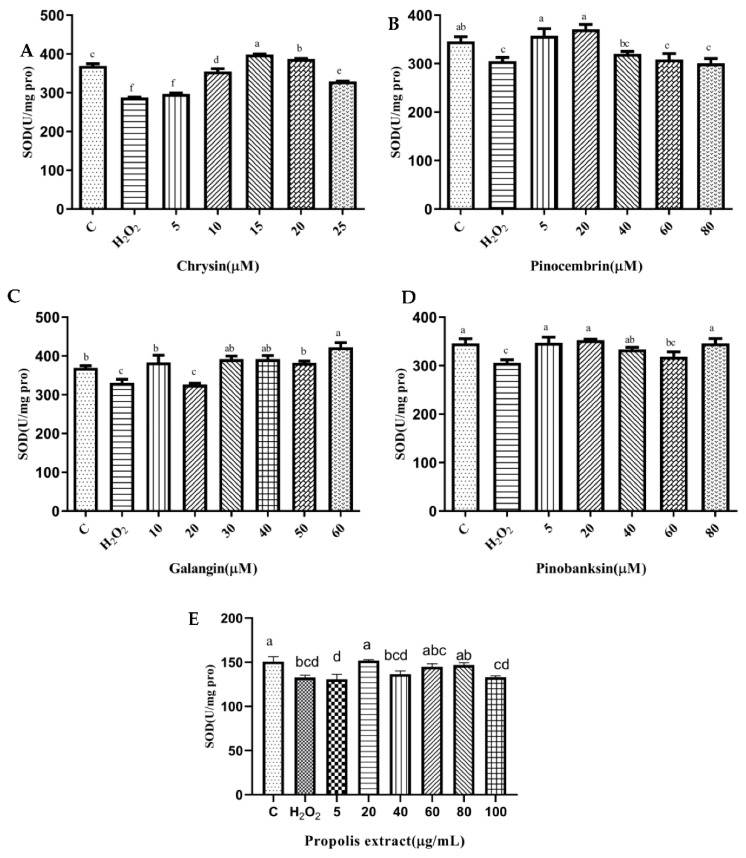
SOD levels induced by individual flavonoids (chrysin, pinocembrin, galangin, and pinobanksin) and propolis extract in 150 μM H_2_O_2_-induced H9c2 cells. H9c2 cells were stimulated with different concentrations of phytochemicals (chrysin, pinocembrin, galangin, pinobanksin, and propolis extract) for 12 h and then treated with 150 μM H_2_O_2_ for 1 h; finally, SOD levels were detected using Total Superoxide Dismutase Assay Kit with WST-8. (**A**) SOD levels in chrysin-induced H9c2 cells. (**B**) SOD levels in pinocembrin-induced H9c2 cells. (**C**) SOD levels in galangin-induced H9c2 cells. (**D**) SOD levels in pinobanksin -induced H9c2 cells. (**E**) SOD levels in propolis-extract-induced H9c2 cells. Values are expressed as the mean ± SEM (*n* = 4). Values with different letters (a, b, c, d, e, f) in the figure showed significant differences at *p* < 0.05.

**Figure 5 foods-11-02439-f005:**
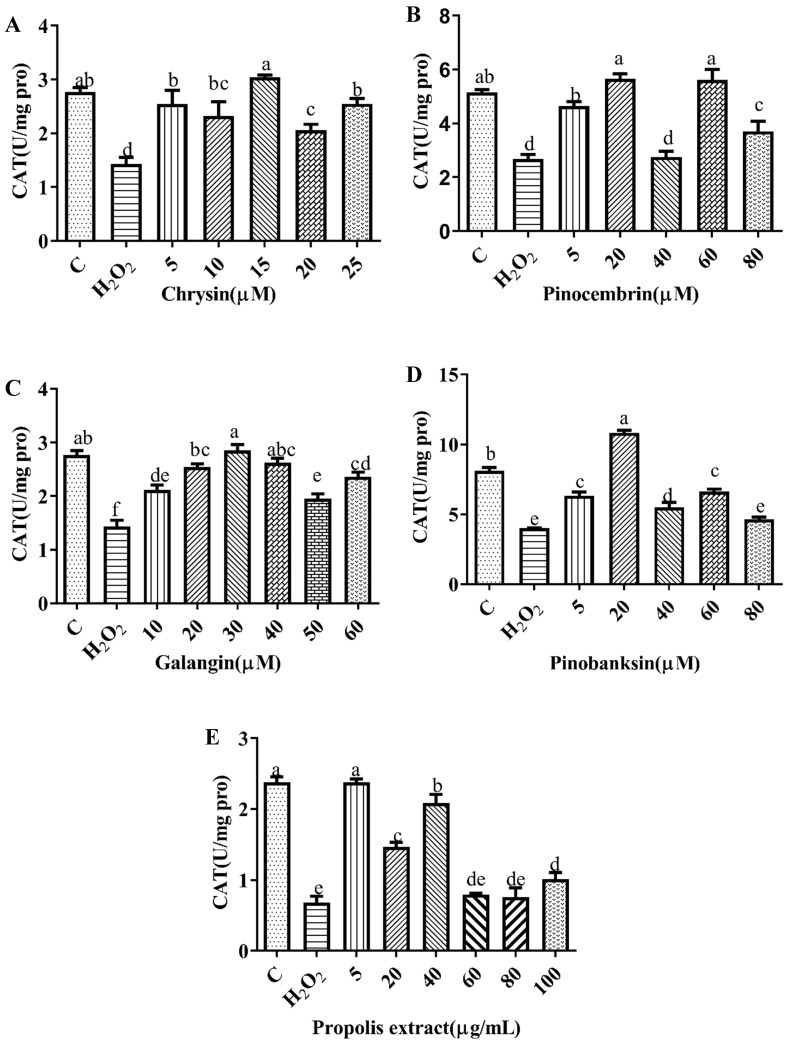
CAT levels induced by individual flavonoids (chrysin, pinocembrin, galangin, and pinobanksin) and propolis extract in 150 μM H_2_O_2_-induced H9c2 cells. H9c2 cells were stimulated with different concentrations of phytochemicals (chrysin, pinocembrin, galangin, and pinobanksin) and propolis extract for 12 h and then treated with 150 μM H_2_O_2_ for 1 h; finally, CAT levels were detected using Catalase Assay Kit. (**A**) CAT levels in chrysin-induced H9c2 cells. (**B**) CAT levels in pinocembrin-induced H9c2 cells. (**C**) CAT levels in galangin-induced H9c2 cells. (**D**) CAT levels in pinobanksin-induced H9c2 cells. (**E**) CAT levels in propolis-extract-induced H9c2 cells. Values are expressed as the mean ± SEM (*n* = 4). Values with different letters (a, b, c, d, e) in the figure showed significant differences at *p* < 0.05.

**Figure 6 foods-11-02439-f006:**
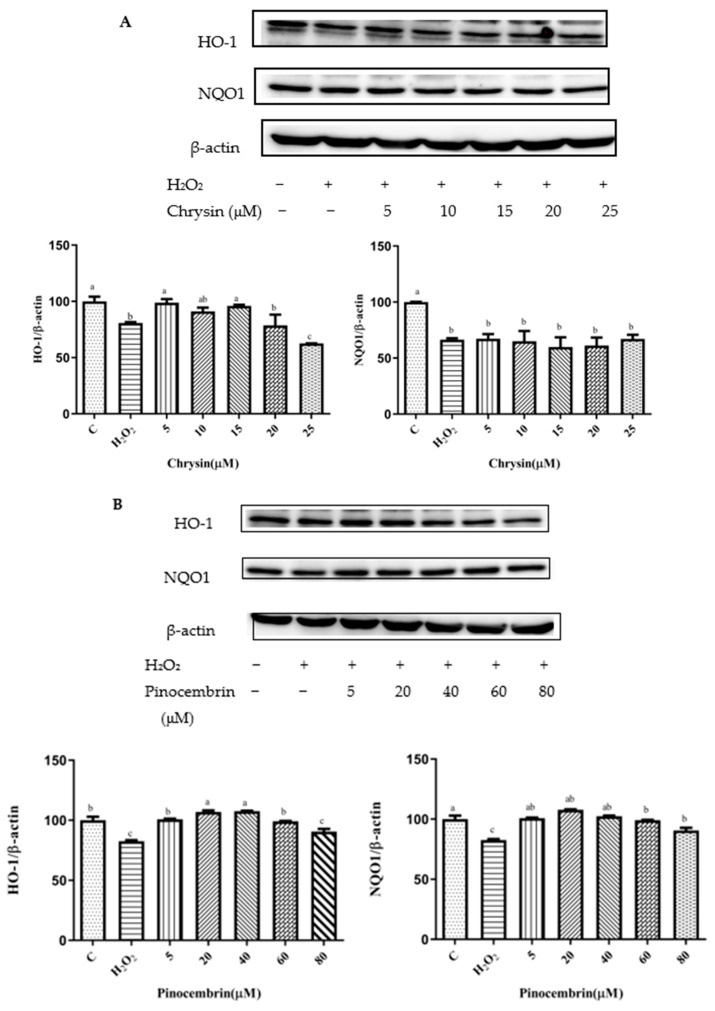
HO-1 and NQO1 expression induced by individual flavonoids (chrysin, pinocembrin, galangin, and pinobanksin) and propolis extract in 150 μM H_2_O_2_-induced H9c2 cells. H9c2 cells were stimulated with different concentrations of phytochemicals (chrysin, pinocembrin, galangin, pinobanksin, and propolis extract) for 12 h and then treated with 150 μM H_2_O_2_ for 1 h; finally, protein expression was detected via Western blot. β-actin was used as an internal control. (**A**) HO-1 and NQO1 expression in chrysin-induced H9c2 cells. (**B**) HO-1 and NQO1 expression in pinocembrin-induced H9c2 cells. (**C**) HO-1 and NQO1 expression in galangin-induced H9c2 cells. (**D**) HO-1 and NQO1 expression in pinobanksin-induced H9c2 cells. (**E**) HO-1 and NQO1 expression in propolis-extract-induced H9c2 cells. Values are expressed as the mean ± SEM (*n* = 3). Values with different (a, b, c, d, e, f, g) letters in the figure showed significant differences at *p* < 0.05.

**Figure 7 foods-11-02439-f007:**
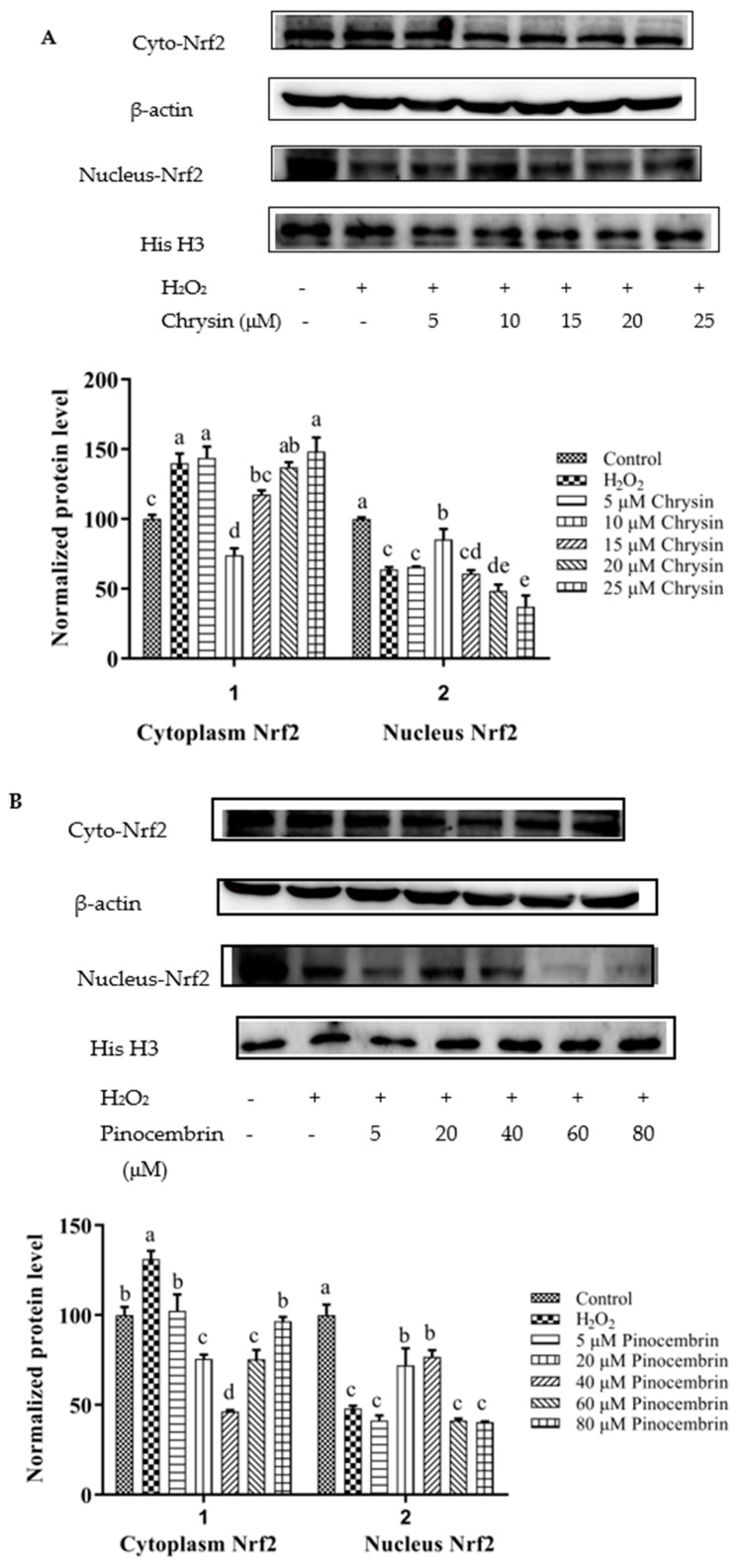
Expression of Nrf2 nuclear translocation induced by individual flavonoids (chrysin, pinocembrin, galangin, and pinobanksin) and propolis extract in 150 μM H_2_O_2_-induced H9c2 cells. H9c2 cells were stimulated with different concentrations of phytochemicals (chrysin, pinocembrin, galangin, pinobanksin, and propolis extract) for 12 h and then treated with 150 μM H_2_O_2_ for 1 h; finally, protein expression was detected via Western blot. β-actin was used as an internal control. (**A**) Expression of Nrf2 nuclear translocation in chrysin-induced H9c2 cells. (**B**) Expression of Nrf2 nuclear translocation in pinocembrin-induced H9c2 cells. (**C**) Expression of Nrf2 nuclear translocation in galangin-induced H9c2 cells. (**D**) Expression of Nrf2 nuclear translocation in pinobanksin-induced H9c2 cells. (**E**) Expression of Nrf2 nuclear translocation in propolis-extract-induced H9c2 cells. Values are expressed as the mean ± SEM (*n* = 3). Values with different (a, b, c, d, e, f, g) letters in the figure showed significant differences at *p* < 0.05.

**Figure 8 foods-11-02439-f008:**
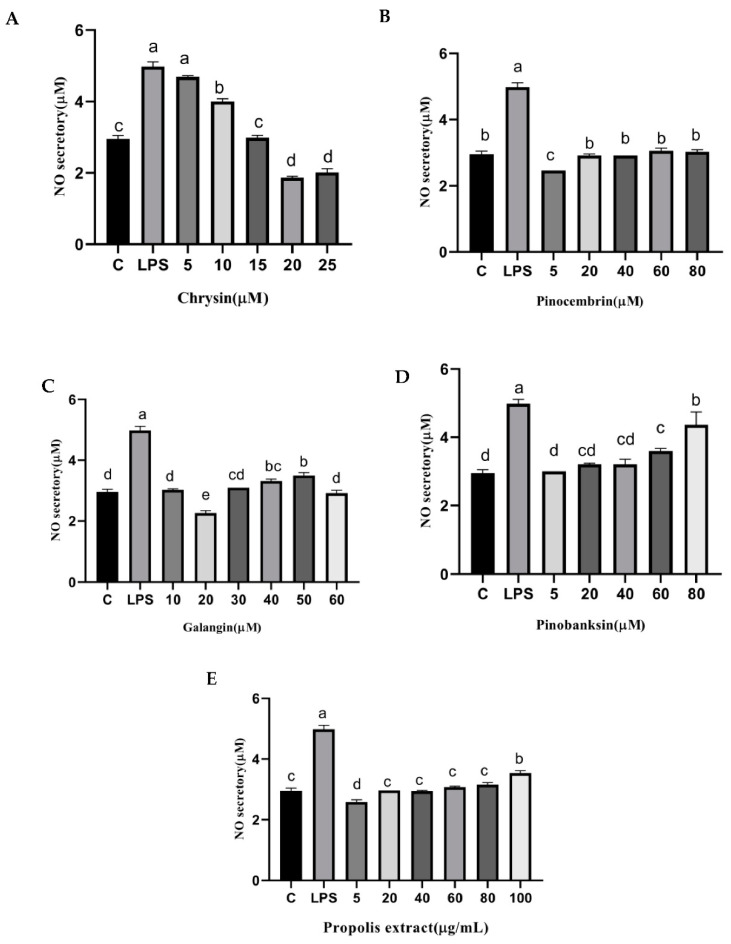
NO levels induced by individual flavonoids (chrysin, pinocembrin, galangin, and pinobanksin) and propolis extract in 10 μg/mL LPS-induced H9c2 cells. H9c2 cells were stimulated with different concentrations of phytochemicals (chrysin, pinocembrin, galangin, pinobanksin, and propolis extract) for 12 h and then treated with 10 μg/mL LPS for 12 h; finally, NO levels were detected with NO Assay Kit. (**A**) NO levels in chrysin-induced H9c2 cells. (**B**) NO levels in pinocembrin-induced H9c2 cells. (**C**) NO levels in galangin-induced H9c2 cells. (**D**) NO levels in pinobanksin-induced H9c2 cells. (**E**) NO levels in propolis-extract-induced H9c2 cells. Values are expressed as the mean ± SEM (*n* = 3). Values with different letters (a, b, c, d, e) in the figure showed significant differences at *p* < 0.05.

**Figure 9 foods-11-02439-f009:**
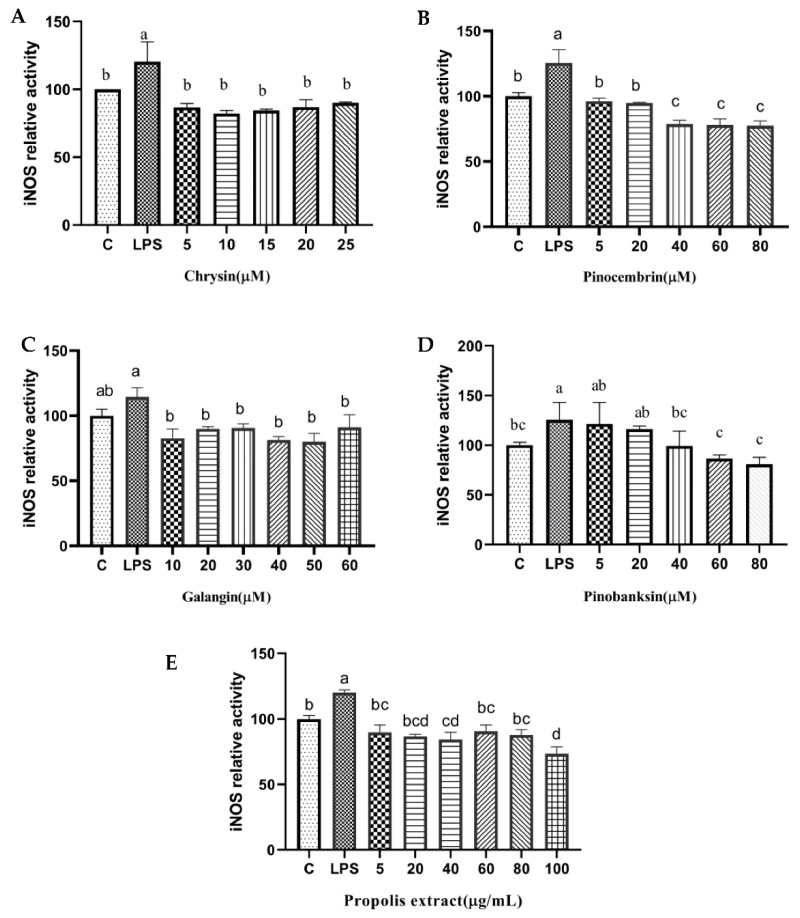
iNOS levels induced by individual flavonoids (chrysin, pinocembrin, galangin, and pinobanksin) and propolis extract in 10 μg/mL LPS-induced H9c2 cells. H9c2 cells were stimulated with different concentrations of phytochemicals (chrysin, pinocembrin, galangin, pinobanksin, and propolis extract) for 12 h and then treated with 10 μg/mL LPS for 12 h; finally, iNOS levels were detected with iNOS Assay Kit. (**A**) iNOS levels in chrysin-induced H9c2 cells. (**B**) iNOS levels in pinocembrin-induced H9c2 cells. (**C**) iNOS levels in galangin-induced H9c2 cells. (**D**) iNOS levels in pinobanksin-induced H9c2 cells. (**E**) iNOS levels in propolis-extract-induced H9c2 cells. Values are expressed as the mean ± SEM (*n* = 3). Values with different letters (a, b, c, d) in the figure showed significant differences at *p* < 0.05.

**Figure 10 foods-11-02439-f010:**
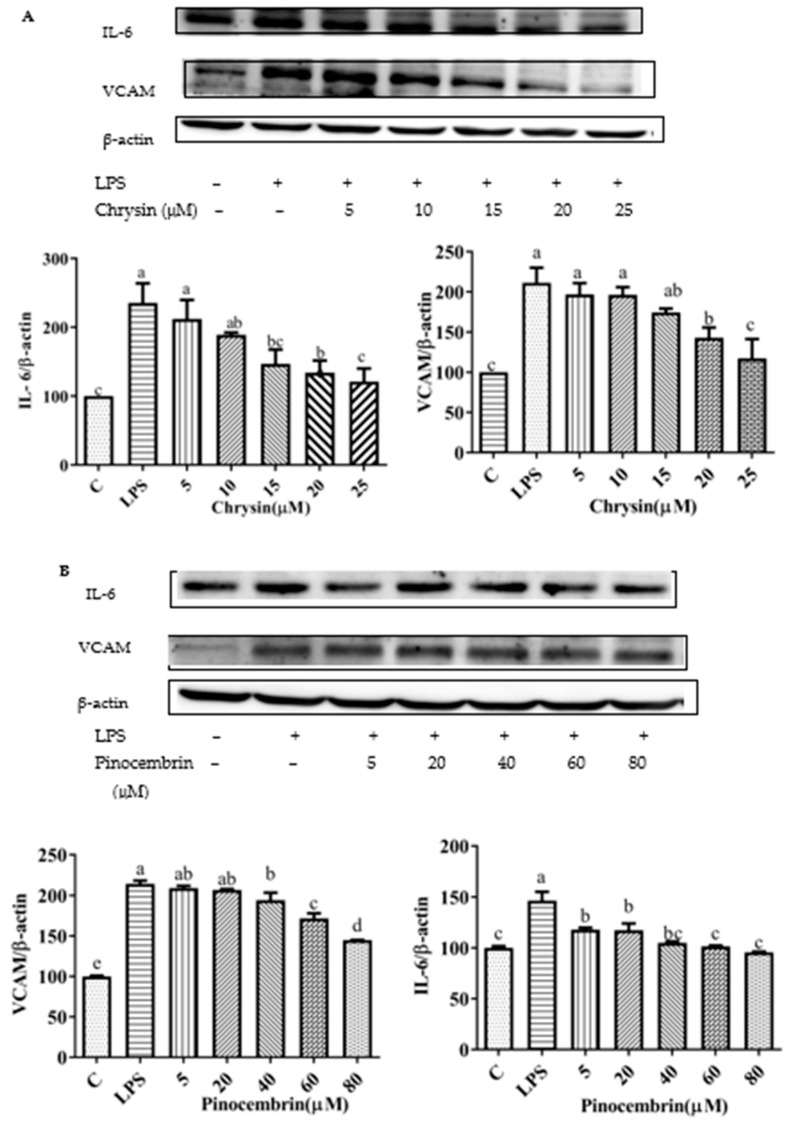
Expression of pro-inflammatory cytokine proteins (VCAM1 and IL-6) induced by individual flavonoids (chrysin, pinocembrin, galangin, and pinobanksin) and propolis extract in 10 μg/mL LPS-induced H9c2 cells. H9c2 cells were stimulated with different concentrations of phytochemicals (chrysin, pinocembrin, galangin, pinobanksin, and propolis extract) for 12 h and then treated with 10 μg/mL LPS for 12 h; finally, the expression of pro-inflammatory cytokine proteins (VCAM1 and IL-6) was detected via Western blot. (**A**) Expression of pro-inflammatory cytokine proteins (VCAM1 and IL-6) in chrysin-induced H9c2 cells. (**B**) Expression of pro-inflammatory cytokine proteins (VCAM1 and IL-6) in pinocembrin-induced H9c2 cells. (**C**) Expression of pro-inflammatory cytokine proteins (VCAM1 and IL-6) in galangin-induced H9c2 cells. (**D**) Expression of pro-inflammatory cytokine proteins (VCAM1 and IL-6) in pinobanksin-induced H9c2 cells. (**E**) Expression of pro-inflammatory cytokine proteins (VCAM1 and IL-6) in propolis-extract-induced H9c2 cells. Values are expressed as the mean ± SEM (*n* = 3). Values with different letters (a, b, c, d, e, f) in the figure showed significant differences at *p* < 0.05.

**Figure 11 foods-11-02439-f011:**
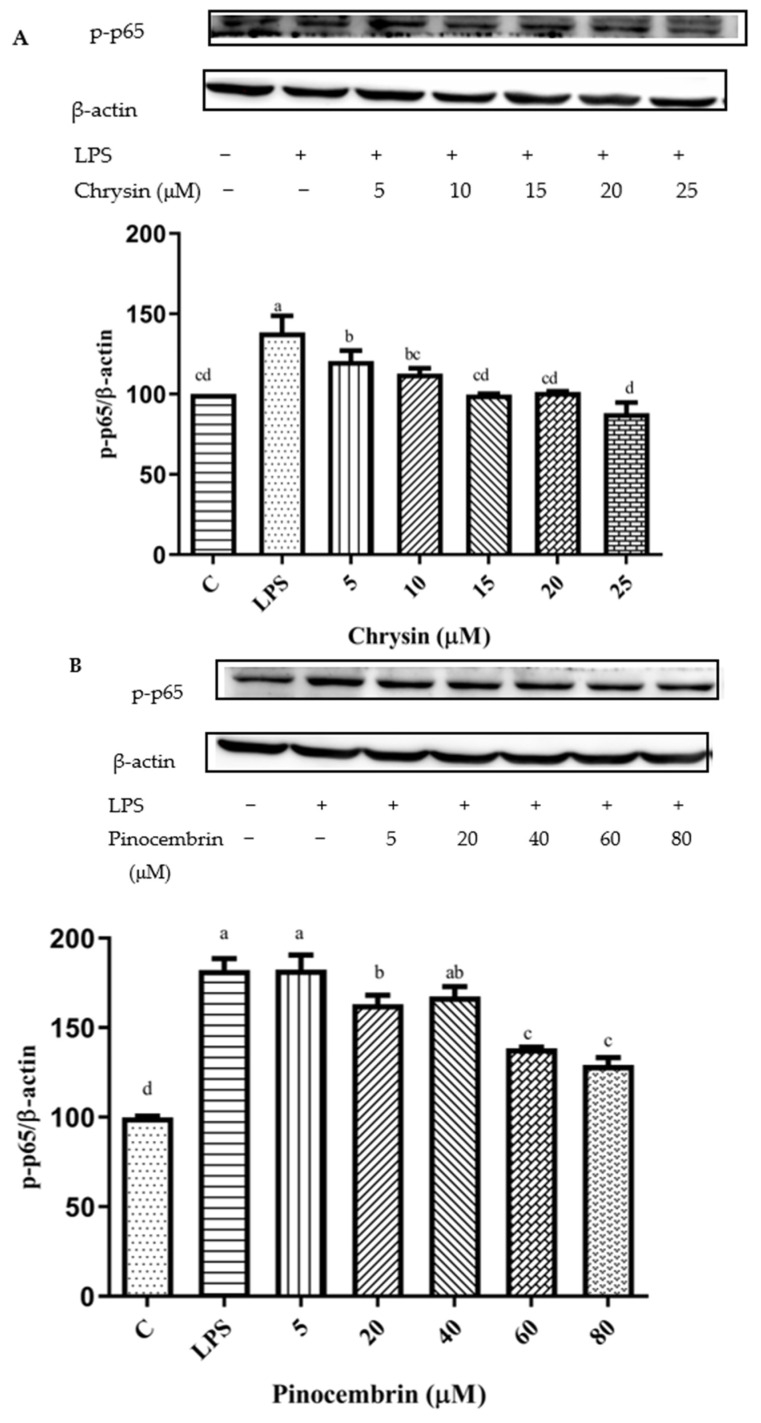
Activation of NF-κB signaling pathway induced by individual flavonoids (chrysin, pinocembrin, galangin, and pinobanksin) and propolis extract in 10 μg/mL LPS-induced H9c2 cells. H9c2 cells were stimulated with different concentrations of phytochemicals (chrysin, pinocembrin, galangin, pinobanksin, and propolis extract) for 12 h and then treated with 10 μg/mL LPS for 12 h; finally, the activation of the NF-κB signaling pathway was detected via Western blot. (**A**) Activation of NF-κB signaling pathway in chrysin-induced H9c2 cells. (**B**) Activation of NF-κB signaling pathway in pinocembrin-induced H9c2 cells. (**C**) Activation of NF-κB signaling pathway in galangin-induced H9c2 cells. (**D**) Activation of NF-κB signaling pathway in pinobanksin-induced H9c2 cells. (**E**) Activation of NF-κB signaling pathway in propolis-extract-induced H9c2 cells. Values are expressed as the mean ± SEM (*n* = 3). Values with different letters (a, b, c, d, e, f) in the figure showed significant differences at *p* < 0.05.

**Table 1 foods-11-02439-t001:** Flavonoids in propolis extract determined via HPLC-TOF-MS analyses.

No.	Identification	RT (min)	Formula	[M-H]/(*m*/*z*)		Major Fragment Ions (m.z)
				Measured	Calculated	
5	Pinobanksin-5-methylether	63.98	C_16_H_14_O_5_	285.0689	285.0692	[M-H-H_2_O]^−^ = 267.0559 [M-H-H_2_O-CH_3_]^−^ = 252.0333 [M-H-H_2_O-CH_3_-CO]^−^ = 224.0397 [^1,3^A]^−^ = 195.0382 [^1,4^A]^−^ =138.0270
6	apigenin-7-O-glucoside	67.33	C_21_H_20_O_10_	431.0854	431.0849	[M-H-C_6_H_11_O_5_]^−^ = 268.0295
7	pinobanksin	68.98	C_15_H_12_O_5_	271.0612	271.0552	[M-H-H_2_O]^−^ = 253.0428 [M-H-H_2_O-CO]^−^ = 225.0489 [M-H-H_2_O-2CO]^−^ =197.0547 [^1,3^A]^−^ = 150.9997 [^1,4^A]^−^ = 125.0218
8	methoxyl-cyanidenon	88.81	C_16_H_12_O_6_	299.0464	299.0464	[M-H-CO_2_]^−^ = 255.0218 [M-H-CH_3_]^−^ = 284.0228 [M-H-CO_2_-CO]^−^ = 227.0276
11	Pinocembrin	97.01	C_15_H_12_O_4_	255.0663	255.0582	[M-H-C_2_H_2_O]^−^ = 213.0472 [M-H-C_3_O_2_]^−^ = 187.0691 [^1,3^A]^−^ = 150.9974 [^1,3^A-CO_2_]^−^ = 107.0091
12	Pinobanksin 3-O-acetate	100.02	C_17_H_14_O_6_	313.0612	313.0609	[M-acetate]^−^ = 271.0509 [M-acetate-H_2_O]^−^ = 253.0419 [M-acetate-H_2_O-CO_2_]^−^ = 209.0529 [M-acetate-H_2_O-C_3_O_2_-C_2_H_2_O]^−^ = 143.0457
14	chrysin	107.44	C_15_H_10_O_4_	253.0419	253.0424	[M-H-CO_2_]^−^ = 209.0517 [M-H-CO_2_-CO]^−^ = 181.0579 [^1,3^A]^−^ = 143.0441 [^1,4^A]^−^ = 107.0000
15	galangin	110.87	C_15_H_10_O_5_	269.0418	269.0423	[M-H-2CO]^−^ = 213.0504 [M-H-2CO-CO_2_]^−^ = 169.0611
16	galangin- 5-methylether	114.38	C_16_H_12_O_5_	283.052	283.0519	[M-H-CH_3_]^−^ = 268.0273 [M-H-CO_2_]^−^ = 239.0262 [M-H-CO_2_-CO]^−^ = 211.0319
17	pinocembrin-7-methylether	123.22	C_16_H_14_O_4_	269.073	269.0729	[M-H-CH_3_]^−^ =254.0485 [M-H-CH_3_-CO]^−^ = 226.0548 [M-H-C_8_H_8_]^−^ = 165.0132 [M-H-C_8_H_8_-CO_2_]^−^ = 121.996
18	Pinobanksin-3-O-butyric	126.72	C_19_H_18_O_6_	341.0917	341.092	[M-butyrate-H_2_O]^−^ = 253.0401

## Data Availability

Data are contained within the article.

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
