# Peer review of "The Antioxidant and Anti-Inflammatory Effects of Flavonoids from Propolis via Nrf2 and NF-κB Pathways"

_foods, 2022, doi:10.3390/foods11162439_

Round 1

Reviewer 1 Report

In this work Xu and coworkers present the antioxidant and anti-inflammatory effects of flavonoids from propolis in in vitro cell based studies. The manuscript is very long and present tremendous work performed by the Authors – congratulations!  The methods are presented with many details and can be repeated, but sometimes not everything is clear. The manuscript is potentially interesting for the readers, but it needs to be revised by the Authors:

  • It seems that the Authors are from the same university, therefore the numbers of affiliation need to be corrected
  • In the abstract I suggest not to present the data in the sentence “For example, chrysin decreased ROS level (243.38+1.22%), increased SOD (298.56+1.59 U/mg protein) and CAT (3.04+0.05 23 U/mg protein) activities, and the expression of HO-1 protein (95.31+1.02%)” but rather the conclusions.
  • The references style needs to be changed according to the journal requirements;
  • Lines 71-75 – the sentences seem to need to be matched together;
  • Line 106 – please, describe the origin of H9c2 cells, as well as explain the usage of this cell line in studies;
  • Please, describe the source of propolis powder;
  • What was the concentration of H2O2 used in the assay 2.5. The production of reactive oxygen species (ROS); explain why cells were digested with 0.02% EDTA- usually with this probe there is direct measurement of fluorescence with the microplate reader; please explain the usage of H2O2 as the main cells stressor;
  • What is the composition of lysis solution mentioned in 2.6. SOD and CAT assays
  • The LPS concentration used in 2.7. NO and NOS assays is relatively high (10 μg/mL); was not LPS cytotoxic to the cells? Why “The supernatant of H9c2 cells was collected after lysis with a tissue lysis solution”, since usually with Griess reagent the NO released to the medium is measured; what is the formula of tissue lysis solution;
  • Describe the detailed procedure how the nucleus Nrf2 was checked and distinguished from cytoNrf2 by western blot;
  • What is the concentration of compounds identified in propolis in part 3.1. Identification of chemical constituents in propolis extract; are these concentrations correlated with studied concentrations for chemically pure compounds identified in propolis? There is no comment in this regard in Conclusions; Please, enlarge the DAD chromatogram at 280 nm of propolis (figure 1)
  • In all figures captions add information of n (number of experiments);
  • In Figure S2 instead of “relative expression of ROS” I suggest “Relative ROS level”
  • Since the manuscript is very long I suggest to present the results in much more shorter way, i.e. do not mention all studied concentrations but rather present the range;
  • Figure 4 – it seems that the numbers present the concentration of flavonoids; what is the H2O2 concentration used in experiment? Similar comment to Figure 5, 6, 7, 9; in the text there are no comments to results presented in all Figures for propolis extract;
  • Figures with western blots need better presentation in case of western blots;
  • Title 3.7 should be modified, since there are results from western blotting – “… on the expression of proteins encoded by Nrf2-downstream antioxidant genes… “ rather than “the expression of Nrf2-downstream antioxidant genes”;
  • Please unify units and present them as μM and not as μmol/L; in case of extract there is often “¦Ìg/mL” – please modify it!
  • Figure 7 – could it be “normalized protein level” instead of “Relative protein expression (% of control)”
  • Figure 8 – in caption add information of LPS concentration;
  • Figure 9C – there is “cell viability” presented instead of “iNOS level”
  • Figure 10 – there are presented many concentrations of LPS, but according to the methodology only one was used (10 µg/mL)
  • Please, in conclusions comment the bioavailability issue of studied compounds and extract – it will greatly enrich presented manuscript.

In summary, I suggest the major revision.

Author Response

Dear professor,

Thanks very much for taking your time to review our manuscript entitled “The antioxidant and anti-inflammatory effects of flavonoids from propolis via Nrf2 and NF-κB pathways” (ID: foods-1734436). I would like to thank you and the reviewers for your suggestions and comments. I have carefully incorporated all comments of you and reviewers into the newly revised manuscript. All major changes are in red in this revised version.

My responses to the reviewers’ comments are below. I believe all their concerns have been addressed, and the quality of the revised manuscript has been improved significantly. I sincerely hope the revised manuscript will be acceptable to you and the reviewers. If you have any further questions about the manuscript, please feel free to contact me. Your consideration for this manuscript to be published in Foods is much appreciated.

Thank you very much for your patient examination. We tried our best to improve the manuscript and make some changes accordingly. We appreciate Editors/Reviewers’ warm work earnestly and hope that the correction will meet with approval. Once again, thank you very much for your comments and suggestions.

Best Regards

Yours sincerely

Hongyan Li

Reviewer 2 Report

The authors investigated anti-oxidative and anti-inflammatory effects of several flavonoids from propolis.

Although the authors performed a lot of work, a presented a lot of results, there are some major issues that are and obstacle for the acceptance.

1. Results obtained in the control and H202-treated cells are not consistent:

·         In Fig. S2 H202 increased ROS below 300% and with small standard error mean (SEM), whereas in Figure 3 H2O2-induced ROS increases were more pronounced, and out of the SEM represented in Fig. S2 (in galangin-group H202-induced increase was 1458%!!!) – please comment these discrepancies. There must be some methodological issue if effect of H2O2 was not consistent.

·         Figure 4 – catalase activity in control group vary too much – assay is not reproducible, conclusions are questionable, almost random effects that are not dose-dependent

·         Figure 6 – various effects of H2O2 on HO-1 and NQO1 expression despite the very small SEMs indicated in each Figure

·         Fig 11 – effect of LPS on p-p65 are very  variable between different blots

2. For most flavonoids, effects are not dose-dependent, they are almost random. It is hard to made any conclusion from such a data. Even for H202, the observed effects are not dose dependent.

·         L314 – „The ROS level increased gradually when the concentration of H2O2 was in the range of 0-150 μM. The ROS level decreased gradually when the  concentration was above 150 μM (p < 0.05).“ – please comment this finding, a dose-response in ROS production probably should be expected for H2O2

·         How drop in ROS, observed only at 10 μM chrysin, could be explained? Considering the very unusual dose-response obtained with chrysin, it is highly recommended to use additional concentrations in the range between 5-15 µM.

3. In all figures letters of significance are not clearly explained (what is compared, level of significance).

4. In several cases, Figures and text do not match. Some Figures should be additionally arranged.

·        Fig.10 is not arranged correctly, it is impossible to see all blots, WB and graphs for pinocembrin, galangin and IL-6 (B,C) do not match

·        L651 – „It was also indicated that Nrf2 translocation from the cytoplasm to the nucleus was up-regulated and  then held relatively constant by the treatment with 10-60 μM galangin  
and 5-80 μM pinobanksin.“
– what is meant by relatively constant, some letters of significance are indicated

·        L645 – „as mediated through the modulation of Nrf2 nuclear translocation. It indicated that Nrf2 translocation from the  cytoplasm to the nucleus was up-regulated (chrysin below 10 μM, pinocembrin below 40 μM and propolis extract below 40 μg/mL) and
then down-regulated (chrysin above 10 μM, pinocembrin above 40 μM
and propolis extract above 40 μg/mL) by the treatment of chrysin,
pinocembrin and propolis extract.“ 
- not correct for pinocembrin

·        L778 – „For example, compared to the control group
(100%), the expression of p-p65 protein was significantly enhanced in  
the LPS-induced group (235.48+28.32%). The expression of p65 protein
was decreased to 212.11+27.49% when the concentration of chrysin was  
at 5 μM. The expression of p65 protein gradually decreased to  
121.05+18.9% when the concentration of chrysin was gradually  
increased to 25 μM.“
– numbers do not match with the graph

5. Standard error means are very small – with such small SEMs, values obtained in control and H2O2-treated cells should be almost identical

·        Line 662 –“ And the level of NO was significantly decreased
(1.866+0.03 μM-4.69+0.03 μM) after 5-25 μM chrysin pretreatment in
H9c2 cells induced by 10 μ g/mL LPS.”

·        The authors wrote that „As for chrysin, compared with the H2O2-induced group (373.35+2.42%), chrysin at concentrations of 5 μM (361.15+1.57%) and 10 μM (243.38+1.22%), the ROS generation decreased.“  I doubt that such small SEMs could be obtained if ROS levels were determined in 3 independent experiments. Please comment the number of experiments performed.

6. Contradictory conclusions

·        L884 – „Secondly, antioxidant or pro-oxidant effects were determined by
the the concentrations of flavonoid. In our study, chrysin, pinocembrin,  
galangin, pinobanksin and propolis extract exhibited antioxidant and  
pro-oxidant effects in a dose-dependent manner.“
– in comparison to H2O2, all examined concentrations of all flavonoids reduced ROS content (antioxidant activity). It should be clearly explained what is meant by pro-oxidative effects

Minor

L56 - instance pathway?

L64 – leukocyteflammation?

L67 – please cite literature relevant for cardiovascular disorders

L90 – „inhibiting intracellular ROS levels and ROS“ – unclear

L94 – SDH – abbreviation?

L131 – Please explain how is propolis powder obtained from the resin

L138 – What supernatants were merged? – unclear

L163 - 10% FBS, 1 U/mL penicillin, and 1 mg/mL streptomycin – should be checked

L183, L193, L213, L233 – „In short, H9c2 cells (1×105 cells per well) were seeded inside a six-well“ – the same density was indicated for 96-well

Table 1 – identification column should be organized better

Figure S3 should be corrected (LPS instead of H2O2)

L656 „NO was produced by inflammatory cells“ – H9c2 cells are not inflammatory cells?

L767 – „NF-κB signaling pathway could be activated by the up-regulation
of pro-inflammatory cytokines[43].“ – vice versa?

L868 – „ It could be observed that  chrysin, pinocembrin and pinobanksin are flavonoids with high content in propolis extract, and galangin is a flavonol compound with  high content in propolis extract.“ – unclear, flavonol is type of flavonoid

Author Response

(The authors gave the same response as above.)

Round 2

Reviewer 1 Report

The Authors answered the most of my concerns, but still I have some comments:

-          I suggest to get rid from the Abstract the concentrations of H2O2 and LPS (line 15) – it is not necessary to present them.

-          Information about usage of H9c2 cells presented in lines 166-176 should be added to the introduction directly after the sentence “Therefore, the propolis extract and their main flavonoids were 100 used to explore the basic mechanisms of antioxidant and anti-101 inflammatory effects in H9c2 cells.”

-          I understand the explanation of chemical identification of propolis compounds. Still, I strongly suggest to add column to the Table 1 presenting the quantitative data for each  identified compound.

After following these recommendation the manuscript may be published.

Author Response

Dear professor,

Thanks very much for taking your time to review our manuscript entitled “The antioxidant and anti-inflammatory effects of flavonoids from propolis via Nrf2 and NF-κB pathways” (ID: foods-1734436). I would like to thank you and the reviewers for your suggestions and comments. I have carefully incorporated all comments of you and reviewers into the newly revised manuscript. All major changes are in red in this revised version.

My responses to the reviewers’ comments are below. I believe all their concerns have been addressed, and the quality of the revised manuscript has been improved significantly. I sincerely hope the revised manuscript will be acceptable to you and the reviewers. If you have any further questions about the manuscript, please feel free to contact me. Your consideration for this manuscript to be published in Foods is much appreciated.

Thank you very much for your patient examination. We tried our best to improve the manuscript and make some changes accordingly. We appreciate Editors/Reviewers’ warm work earnestly and hope that the correction will meet with approval. Once again, thank you very much for your comments and suggestions.

Best Regards

Yours sincerely

Hongyan Li

The Authors answered the most of my concerns, but still I have some comments:

-          I suggest to get rid from the Abstract the concentrations of H2O2 and LPS (line 15) – it is not necessary to present them.

Response: Thank you very much for your comments. The sentence has been deleted accordingly in the revised version.

-          Information about usage of H9c2 cells presented in lines 166-176 should be added to the introduction directly after the sentence “Therefore, the propolis extract and their main flavonoids were 100 used to explore the basic mechanisms of antioxidant and anti-101 inflammatory effects in H9c2 cells.”

Response: Thank you. The information has been revised as you suggested in the revised version.

-          I understand the explanation of chemical identification of propolis compounds. Still, I strongly suggest to add column to the Table 1 presenting the quantitative data for each  identified compound.

Response: Thank you very much for your patience. Actually, we have presented the quantitative data of the four main phytochemicals in our previous paper (Reference 38: J Food Sci. 2019, 84(12), 3850-3865). Therefore, the quantitative data was directly cited and used in this manuscript. However, due to the lack of commercial standards, not all of the compounds were quantitated. In Line 1017-1020, we have added “According to our previous data, the main flavonoids of the propolis extract were identified to chrysin (30.56 ± 0.60 mg/g DW), pinocembrin (30.96 ± 0.34 mg/g DW) and galangin (16.69 ± 0.45 mg/g DW) and pinobanksin (9.36 ± 0.28 mg/g DW) ” in the revised version. 

Reviewer 2 Report

In my opinion, this manuscript should not be accepted.

It is very hard to see any pattern in the results. The main conclusion „Moreover, results revealed that phytochemicals presented antioxidant effects at lower concentrations, while presented pro-oxidant effects and anti-inflammatory effects at higher concentrationsis not visible. For example, expressions of IL-6 and VCAM are dose-dependently reduced  by all investigated compounds. It seems to me that lower concentrations also demonstrate anti-inflammatory activity. Expression of HO-1 for galangin and pinobanksin indicates only anti-inflammatory activity….

Statistics is not clear (explanation of the letters a-e is not provided).

The overall effect on the viability in the presence of hydrogen peroxide and flavonoids should be demonstrated.

Author Response

Dear professor,

Thanks very much for taking your time to review our manuscript entitled “The antioxidant and anti-inflammatory effects of flavonoids from propolis via Nrf2 and NF-κB pathways” (ID: foods-1734436). I would like to thank you and the reviewers for your suggestions and comments. I have carefully incorporated all comments of you and reviewers into the newly revised manuscript. All major changes are in red in this revised version.

My responses to the reviewers’ comments are below. I believe all their concerns have been addressed, and the quality of the revised manuscript has been improved significantly. I sincerely hope the revised manuscript will be acceptable to you and the reviewers. If you have any further questions about the manuscript, please feel free to contact me. Your consideration for this manuscript to be published in Foods is much appreciated.

Thank you very much for your patient examination. We tried our best to improve the manuscript and make some changes accordingly. We appreciate Editors/Reviewers’ warm work earnestly and hope that the correction will meet with approval. Once again, thank you very much for your comments and suggestions.

Best Regards

Yours sincerely

Hongyan Li

In my opinion, this manuscript should not be accepted.

It is very hard to see any pattern in the results. The main conclusion „Moreover, results revealed that phytochemicals presented antioxidant effects at lower concentrations, while presented pro-oxidant effects and anti-inflammatory effects at higher concentrations“ is not visible. For example, expressions of IL-6 and VCAM are dose-dependently reduced  by all investigated compounds. It seems to me that lower concentrations also demonstrate anti-inflammatory activity. Expression of HO-1 for galangin and pinobanksin indicates only anti-inflammatory activity….

Response: Thank you very much for your comments. We have made changes in the revised manuscript to make the result more clear. The main conclusion was "For chrysin, it showed anti-oxidant effects at concentration range 5 μM to 10 μM, while showed pro-oxidant effect for concentration range 15 μM to 25 μM, and showed anti-inflammatory effects for concentration range 5 μM to 25 μM. For pinocembrin, it showed anti-oxidant effects at concentration range 5 μM to 40 μM, while showed pro-oxidant effect for concentration range 60 μM to 80 μM, and showed anti-inflammatory effects for concentration range 5 μM to 80 μM. For galangin, it showed anti-oxidant effects at concentration range 10 μM to 50 μM, while showed pro-oxidant effect for concentration range 50 μM to 60 μM, and showed anti-inflammatory effects for concentration range 10 μM to 60 μM. For pinobanksin, it showed anti-oxidant effects at concentration range 5 μM to 60 μM, while showed pro-oxidant effect for concentration range 60 μM to 80 μM, and showed anti-inflammatory effects for concentration range 5 μM to 80 μM. For propolis extract, it showed anti-oxidant effects at concentration range 5 μg/mL to 40 μg/mL, while showed pro-oxidant effect for concentration range 60 μg/mL to 100 μg/mL, and showed anti-inflammatory effects for concentration range 5 μg/mL to 100 μg/mL. What’s more, the higher concentration compound was, the stronger anti-inflammatory was showed” (Line 693-682).

The modifications were listed as follows:

Line 392-412: For chrysin (Fig. 3A), compared with the H2O2-induced group (373.35±2.42%), it showed anti-oxidant effects at concentration range 5 μM (361.15±1.57%) to 10 μM (243.38±1.22%), while showed pro-oxidant effect for concentration range 15 μM (331.01±4.16%) to 25 μM (354.88±1.58%). For pinocembrin (Fig. 3B), compared with the H2O2-induced group (373.64±17.37%), it showed anti-oxidant effects at concentration range 5 μM (294.56±7.6%) to 40 μM (179.94±6.2%), while showed pro-oxidant effect for concentration range 60 μM to 80 μM. For galangin (Fig. 3C), compared with the H2O2-induced group (430±1%), it showed anti-oxidant effects at concentration range 10 μM (384.59±0.87%) to 50 μM (232.27±6.69%), while while showed pro-oxidant effect for concentration range 50 μM to 60 μM (240.7±2.33%). For pinobanksin (Fig. 3D), compared with H2O2-induced group (461.88±36.48%), it showed anti-oxidant effects at concentration range 5 μM (320.97±14.04%) to 40 μM (179.94±6.2%), while showed pro-oxidant effect for concentration range 40 μM to 80 μM (182.81+4.3%). For propolis extract (Fig. 3E), compared with the H2O2-induced group (331.87±4.56%), it showed anti-oxidant effects at concentration range 10 μg/mL (236.43±3.16%) to 40 μg/mL (49.78±2.6%), while showed pro-oxidant effect for concentration range 60 μg/mL (112.79±1.58%) to 100 μg/mL (223.64±3.17%).

Line 526-532: For example, for chrysin (Fig. 6A), compared with the H2O2-induced group (80.73±0.88%), the expression of HO-1 was significantly increased and antioxidant effect increased at concentration range 5 μM (98.85±3.22%) to 15 μM (95.31±1.02%), while the expression of HO-1 was gradually decreased and it showed pro-oxidant effect for concentration range 20 μM (78.69±9.44%) to 25 μM (62.59±0.06%).

Line 534-540: For example, for pinobanksin (Fig.6D), compared with the H2O2-induced group (46.7±0.74%), the expression of HO-1 was significantly increased and antioxidant effect increased at concentration range 5 μM (75.13±0.83%) to 40 μM (96.85±3.82%), while the expression of HO-1 was no longer consistently elevated and it showed pro-oxidant effect for concentration range 40 μM to 80 μM (105.95±4%).

Line 545-551: For example, for propolis extract ((Fig.6E), compared with the H2O2-induced group (60.36±0.38%), the expression of NQO1 was significantly increased and antioxidant effect increased at concentration range 5 μM (90.88±4.4%) to 40 μM (106.87±7.5%), while the expression of NQO1 protein was no longer consistently elevated and it showed pro-oxidant effect for concentration range 40 μM to 100 μM (114.55±1.97%).

Line 668-674: For example, for chrysin (Fig. 7A), compared with the H2O2-induced group, the translocation of Nrf2 from the cytoplasm to the nucleus was gradually increased and showed anti-oxidant effects at concentration range 5 to 10 μM, while the translocation of Nrf2 from the cytoplasm to the nucleus was gradually decreased in a dose-dependent manner and it showed pro-oxidant effects for concentration range 15 μM to 25 μM.

Line 679-684: For example, for galangin, compared with the H2O2-induced group, the translocation of Nrf2 from the cytoplasm to the nucleus was increased in a dose-dependent manner and showed anti-oxidant effects at concentration range 10 μM to 50 μM, while the translocation of Nrf2 from the cytoplasm to the nucleus might have reached saturation and it showed pro- oxidant effects for concentration range 50 μM to 60 μM.

Line 1033-1052: It was revealed that flavonoids from propolis mainly presented antioxidant effects at lower concentrations, and presented pro-oxidant as well as stronger anti-inflammatory effects at higher concentrations. It showed anti-oxidant effects in chrysin at concentration range 5 μM to 10 μM, in pinocembrin at concentration range 5 μM to 40 μM, in galangin at concentration range 10 μM to 50 μM, in pinobanksin at concentration range 5 μM to 60 μM, and in propolis extract at concentration range 5 μg/mL to 40 μg/mL. While it showed pro-oxidant effect in chrysin at concentration range 15 μM to 25 μM, in pinocembrin at concentration range 60 μM to 80 μM, in galangin at concentration range 50 μM to 60 μM, in pinobanksin at concentration range 60 μM to 80 μM, and in propolis extract at concentration range 60 μg/mL to 100 μg/mL. In addition, it showed anti-inflammatory effects in chrysin at concentration range 5 μM to 25 μM, in pinocembrin at concentration range 5 μM to 80 μM, in galangin at concentration range 10 μM to 60 μM, in pinobanksin at concentration range 5 μM to 80 μM, and in propolis extract at concentration range 5 μg/mL to 100 μg/mL. What’s more, the higher concentration compound was, the stronger anti-inflammatory was showed.

Statistics is not clear (explanation of the letters a-e is not provided).

Response: Thank you very much. The Statistics was revised in the newly manuscript. For example, “Values with different letters (a, b, c, d, e) in the figure indicated significant differences at p < 0.05”.

The overall effect on the viability in the presence of hydrogen peroxide and flavonoids should be demonstrated.

Response: Thank you very much for your patience in checking our article. Actually, CCK-8 assay was used for the measurement of cytotoxicity, and the data of CCK-8 assay were used for selecting the concentration of the compounds in further experiment. The ROS level was used to determine the antioxidant activity of the compounds.
